# Nutrient limitations regulate soil greenhouse gas fluxes from tropical forests: evidence from an ecosystem-scale nutrient manipulation experiment in Uganda

Joseph Tamale[1,5], Roman Hüppi[4], Marco Griepentrog[3], Laban Frank Turyagyenda[5], Matti Barthel[4], Sebastian Doetterl[3], Peter Fiener[1*], and Oliver van Straaten[2,6]

[1]Institute of Geography, University of Augsburg, Augsburg, 86159, Germany
[2]Environmental Control Department, Northwest German Forest Research Institute, 37079, Germany
[3]Soil Resources, Department of Environmental Systems Science, ETH, Zurich, 8092, Switzerland
[4]Sustainable Agroecosystems, Department of Environmental Systems Science, ETH, Zurich, 8092, Switzerland
[5]Ngetta Zonal Agricultural Research and Development Institute (NGEZARDI), P. O. Box 52, Lira, Uganda
[6]Soil Science of Tropical and Subtropical Ecosystems, Büsgen-Institute, University of Göttingen, Göttingen, 37077, Germany

*Correspondence to: Peter Fiener (peter.fiener@geo.uni-augsburg.de)

**Abstract.** Soil macronutrient availability is one of the abiotic controls that alters the exchange of greenhouse gases (GHGs) between the soil and atmosphere in tropical forests. However, evidence on macronutrient regulation of soil GHG fluxes from central African tropical forests is still lacking—limiting our understanding of how these biomes could respond to potential future increases in nitrogen (N) and phosphorus (P) deposition. The aim of this study was to disentangle the regulation effect of soil nutrients on soil GHG fluxes from a Ugandan tropical forest reserve in the context of increasing N and P deposition. Therefore, a large-scale nutrient manipulation experiment (NME) based on 40 m x 40 m plots with different nutrient addition treatments (N, P, N + P, and control) was established in Budongo Forest Reserve. Soil carbon dioxide ($CO_2$), methane ($CH_4$), and nitrous oxide ($N_2O$) fluxes were measured monthly using permanently installed static chambers for 14 months. Total soil $CO_2$ fluxes were partitioned into autotrophic and heterotrophic components through a root trenching treatment. In addition, soil temperature, soil water content, and nitrates were measured in parallel to GHG fluxes. N addition (N, N + P) resulted in significantly higher $N_2O$ fluxes in the transitory phase (0-28 days after fertilization, $p < 0.01$), because N fertilization likely increased soil N beyond the microbial immobilization and plant nutritional demands leaving the excess to be nitrified or denitrified. Prolonged N fertilization however, did not elicit a significant response in background (measured more than 28 days after fertilization) $N_2O$ fluxes. P fertilization marginally and significantly increased transitory ($p = 0.05$) and background ($p = 0.01$) $CH_4$ consumption, probably because it enhanced methanotrophic activity. Addition of N and P together (N + P) resulted in larger $CO_2$ fluxes in the transitory phase ($p = 0.01$), suggesting a possible co-limitation of both N and P on soil respiration. Heterotrophic (microbial) $CO_2$ effluxes were significantly higher than the autotrophic (root) $CO_2$ effluxes ($p < 0.01$) across all treatment plots with microbes contributing about two thirds of the total soil $CO_2$ effluxes. However, neither heterotrophic nor autotrophic respiration significantly differed between treatments. The results from this study suggest that the feedback of tropical forests to the global soil GHG budget could be disproportionately altered by increases in N and P availability over these biomes.

## 1    Introduction

Tropical forest soils play an important role in the earth's radiative balance by sequestering and releasing significant amounts of carbon dioxide ($CO_2$), methane ($CH_4$), and nitrous oxide ($N_2O$) (Mosier et al., 2004). It is estimated that tropical forest soils emit about $1.3 \pm 0.3$ Tg $N_2O$ yr$^{-1}$ (Butterbach-Bahl et al., 2004), capture 6.4 Tg $CH_4$ yr$^{-1}$ (Dutaur and Verchot, 2007), and sequester about 10 % of the total atmospheric $CO_2$ via photosynthesis; and account for about 30 % of the world's soil C stocks (Jobbágy and Jackson, 2000; Malhi and Phillips, 2004).

The rate and magnitude of the specific plant- and soil microbial-processes that produce ($CO_2$: autotrophic and heterotrophic respiration, $N_2O$: denitrification and nitrification, $CH_4$: enteric fermentation and methanogenesis) and consume ($CO_2$: photosynthesis, $CH_4$: oxidation) greenhouse gases (GHGs) in and at the soil-atmospheric interface are constrained by a multiplicity of biotic and abiotic controls (Mosier et al., 2004). These controls include vegetation communities (Veber et al., 2018), soil moisture (Sjögersten et al., 2018), soil temperature (Holland et al., 2000), geochemistry (given its control on microbial abundance (Gray et al., 2014) and soil organic carbon stabilization (Doetterl et al., 2015)), as well as macronutrient availability (especially N and P) (Oertel et al., 2016).

Macronutrient replenishment in undisturbed tropical forests is inherently via litter input (for both N and P, (Tanner et al., 1998)) and rock weathering (for P, (Hedin et al., 2003)) processes. However, the past three decades have seen an increase in the levels of N and P deposition over most tropical regions (including central Africa) due to widespread deforestation and biomass burning (Bauters et al., 2019; Galloway et al., 2004). Currently, the central African region receives about 18.5 kg N ha$^{-1}$ (Bauters et al., 2019) and $1.8 - 2.5$ kg P ha$^{-1}$ (Tamatamah et al., 2005) each year due to high fire-derived N deposition (Bauters et al., 2019) and P rich biomass aerosols (Barkley et al., 2019), respectively. Increased anthropogenic N and P deposition over tropical forest biomes disrupts ecosystem stoichiometric equilibrium thereby affecting biogeochemical cycling of N and P (Bauters et al., 2019), as well as, the exchange of GHGs between the soil and atmosphere (Corre et al., 2014). One way of understanding how increases in N and P availability (for instance through deposition) affect soil GHG fluxes from tropical forests is through large scale nutrient manipulation experiments (NMEs). NMEs purposely use large doses of N and P (e.g. Cleveland and Townsend, 2006 (150 kg N ha$^{-1}$ yr$^{-1}$ and 150 kg P ha$^{-1}$ yr$^{-1}$), Hall and Matson, 2003 (100 kg N ha$^{-1}$ yr$^{-1}$ and 40 kg P ha$^{-1}$ yr$^{-1}$)) to simulate how future nutrient enrichment of tropical forests (through deposition) could affect soil GHG fluxes (among other ecosystem processes) (Corre et al., 2010).

To date, several NMEs have been carried out across the tropics (e.g. Corre et al., 2010; Wei et al., 2008), and the outcome has been a consensus that addition of N to an already N-rich tropical forest ecosystem results in increased $N_2O$ emissions (Corre et al., 2014; Martinson et al., 2013; Zhang et al., 2008). For N-rich forest ecosystems, an increase in available soil N beyond the microbial immobilization and plant nutritional demands, results in the excess being nitrified or (and) denitrified by soil microbes (Corre et al., 2014). However, several studies suggest that increased availability of N not only reduces fine root biomass but also curtails microbial activity leading to reduced autotrophic (Cusack et al., 2011) and heterotrophic respiration (Chen et al., 2010; DeForest et al., 2006; Koehler et al., 2009a), respectively. Notably, there are varying results on how N addition affects $CH_4$ uptake from tropical forest soils. For instance, Veldkamp et al. (2013) found no effect of N on $CH_4$ uptake while Du et al. (2019) measured reduced $CH_4$ consumption following addition of N to a tropical forest, with the latter study suggesting an inhibitory effect of N on $CH_4$ uptake (Bodelier and Steenbergh, 2014; Seghers et al., 2003; Zhang et al., 2011). Aronson and Helliker (2010)

argue that the observed differences in the measured $CH_4$ fluxes in the two separate studies were likely due to the different amounts of N added in the respective experimental setups. They argued that low amounts of N stimulate $CH_4$ uptake while high amounts inhibit it.

With respect to P, it has been shown that P availability opens up the N cycle by stimulating soil organic matter mineralization, releasing excess N for soil nitrification or (both) denitrification processes (Mori et al., 2010). It is also urged that P availability has a positive effect on both autotrophic and heterotrophic components of soil respiration (Mori et al., 2013). P not only stimulates fine root growth (Chen et al., 2010) but also regulates organic matter decomposition (Mori et al., 2018). However, studies elucidating P limitation of organic matter decomposition in the P deficient tropics remain rare and even the few available studies on the regulation effect of P on leaf litter mass loss rates are inconclusive (Cleveland and Townsend, 2006). This might explain why contrasting results were reported from two similar experiments carried out on P depleted soils in Hawaii (Hobbie and Vitousek, 2000) and the Brazilian Amazon (McGroddy et al., 2008). Hobbie and Vitousek (2000) reported an increase in litter mass loss rate while McGroddy et al. (2008) did not detect any change, suggesting that the relationship between P availability and organic matter decomposition is complex (Cleveland and Townsend, 2006). Similarly, literature on the interaction between N and P in regulating $CH_4$ fluxes from tropical forests remains limited.

Despite the recognition that N and P affect soil GHG fluxes and the fact that tropical forest ecosystems could subtly respond to potential future increases in N and P deposition (Bobbink et al., 2010; Li et al., 2006), the magnitude and direction of this response remains unclear for African tropical forests. To date, only a handful of nutrient manipulation experiments (NMEs) focusing on tropical forests response to shifts in ecosystem N and P dynamics have been carried out. Of these studies, just a few included both N and P treatments in their experimental setups (e.g. Corre et al., 2014). Yet, P deficiency typical of tropical soils can have direct impacts on ecosystem biomass production if the limitation is lifted (John et al., 2007). Furthermore, nearly all the studies conducted in (sub-) tropical forest ecosystems were so far concentrated in China (Jiang et al., 2016; Yan et al., 2008; Zheng et al., 2016), Central America (Corre et al., 2014; Koehler et al., 2009a; Matson et al., 2014) and South America (Martinson et al., 2013; Müller et al., 2015; Wolf et al., 2011).

Unfortunately, no single controlled experiment has simulated the effects of elevated soil nutrient inputs on soil greenhouse gas fluxes from African tropical forests despite the projected increase in N and P deposition over these biomes (Galloway et al., 2004) and the fact that they represent a significant proportion of global tropical forests (27 % ; Saatchi et al., 2011). It was for this reason that a replicated completely randomized NME was established in a Ugandan tropical forest reserve to investigate the role nitrogen and phosphorus have in regulating soil GHG fluxes in the context of changing N and P deposition rates over the tropics. It was hypothesized that:

1) addition of N or N + P to a tropical forest ecosystem would result in increased $N_2O$ emissions coming from excess availability of bio-available N beyond microbial immobilization and plant N demands, decreased $CH_4$ uptake due to negative effects of N addition on soil methanotrophs, and reduced $CO_2$ effluxes largely attributed to reduction in both root and microbial respiration upon addition of N;

2) adding P to a tropical forest ecosystem would stimulate release of N from soil organic matter and consequently lead to increased $N_2O$ emissions, higher $CO_2$ effluxes linked to increased root activity and decomposition of soil organic matter, and increased $CH_4$ uptake due to stimulation of methanotrophic activity.

## 2 Materials and Methods

### 2.1 Study site description

The study was conducted in Budongo Forest Reserve, a semi-deciduous tropical forest, located in the northwestern part of Uganda (1°44'28.4" N, 31°32'11.0" E). The forest reserve spans over 825 km$^2$ and is extensively diverse in respect to forest communities, with *Cynometra alexandria, Chryophyllum albidum, Meosopsis eminii and Diospyros abyssinica* as the dominant tree species (Eggeling, 1947). The long-term mean annual temperature and precipitation over the study area is 25 °C and 1700 mm, respectively (Lukwago et al., 2020). Rainfall is distributed into two rainy seasons (i.e. March to May and August to November) punctuated by a strong dry season (December to February), and a weak dry season (June to July) (Lukwago et al., 2020). It is worth noting that the amount of rainfall received during the field campaign (2385 mm, Fig. 2d) was higher than the long-term mean annual precipitation for this region. The weather data for the experiment period was obtained from a climatic station installed at Budongo Conservation Field station (2 km northwest of the study site), and was beneficial to understand how precipitation constrained soil greenhouse gas fluxes given its direct control on water filled pore space. The soils at the experimental site are highly weathered—classified as Lixisols (IUSS Working Group WRB, 2014), and are developed on a Precambrian gneissic-granulitic basement complex (van Straaten, 1976).

### 2.2 Experimental design

The study was conducted within the framework of a running nutrient manipulation experiment (NME). The NME study used a completely randomized design to investigate how the three macronutrients (applied individually as nitrogen (N), phosphorus (P), and potassium (K) and in all possible combinations (N + P, N + K, P + K, N + P + K) as treatments) constrained key ecosystem processes (particularly nutrient cycling, and net primary productivity) in comparison to the unamended control. Each of the eight treatments was replicated four times (hence, n = 32 plots; 8 treatments x 4 replications). While the NME included a K treatment, the soil GHG flux study—the basis for this manuscript was conducted on the N, P and N + P (combination of N and P) plots, and compared to the untreated control plots (n = 16). Only N and P (among nutrient addition plots) were exclusively considered for soil GHG flux measurements because their availability has been shown to limit soil greenhouse gas fluxes from tropical forest biomes. Each treatment plot measured 40 m x 40 m in size but measurements were conducted in the inner measurement core (30 m x 30 m) to avoid boundary effects. A spacing of at least 40 m between experimental plots was ensured to prevent spillover of applied nutrients from the neighboring plots. In order to elicit an ecosystem response, N was applied at a rate of 125 kg N ha$^{-1}$ yr$^{-1}$ in form of urea ((NH$_2$)$_2$CO), and P at 50 kg P ha$^{-1}$ yr$^{-1}$ as triple superphosphate (Ca(H$_2$PO$_4$)$_2$). The types of fertilizers and application rates used in this study were identical to those used in the Wright et al. (2011) NME. The fertilizer was applied by hand, and in four split dozes every year. Specifically, 31.3 kg N ha$^{-1}$ and 12.5 kg P ha$^{-1}$ were applied to the plots of the NME every three months between May 2018 and June 2020.

## 2.3 Baseline soil physico-biochemical characterization

Prior to the first fertilizer application, soil samples were taken from all the treatment plots of the NME (for the top soils) and from the close proximity of the NME (for deeper soil layers) for baseline soil physico-biochemical analyses. The analyses included; texture, bulk density, soil pH, total soil organic carbon (TOC) stocks, total nitrogen stocks, C/N ratio, exchangeable bases, effective cation exchange capacity (ECEC), and Bray extractable P. For the top soils (0 -10 cm depth), soil monoliths (20 cm (L) x 20 cm (W) x 10 cm (D)) were carefully taken from ten different locations within each plot of the NME (n = 32 plots) using a spade. For deeper soil layers (0 - 30 cm and 30 - 50 cm), samples were obtained outside the established NME plots in order to minimize modifications to the microenvironment inside the NME plots. Deeper soil sampling was done during a reconnaissance survey conducted at approximately 500 m from the current location of the NME site. During the reconnaissance survey, sixteen plots (n = 16) were established and samples taken from five different locations in each plot for every depth interval (i.e. 0 - 30 cm and 30 - 50 cm) using an auger (diameter = 30 mm). The samples from the same depth within each plot were mixed thoroughly in a basin, and about 500 g of the homogenized samples sent to the soils laboratory of the University of Göttingen, Germany, for analysis. Soil texture was determined using a Bouyoucos hydrometer. Soil pH was determined in 1:2.5 (soil water) suspension. Soil bulk density for every depth in each plot was calculated from the mass of oven dried soil (at 105 °C for 48 hours) and the volume of the Kopecky ring (volume = 251 $cm^3$) used in collecting the soil sample. Note that soil bulk density was corrected for stone content. The soils were tested for presence of inorganic carbon (IC) using dilute hydrochloric acid, and were found to be devoid of any IC. Hence, TOC and N were determined using a CN elemental analyzer (Vario EL Cube, Elementar Analysis Systems GmbH, Hanau, Germany) and stocks later calculated from bulk density measurements. Exchangeable base cations (Ca, Mg, K, Na, Al) and ECEC were determined on the 1 - 2 mm earth fraction of the collected soil samples.

## 2.4 Soil greenhouse gas fluxes and soil environmental control measurements

Soil $CO_2$, $CH_4$ and $N_2O$ fluxes were measured monthly over a period of fourteen months (May 2019 to June 2020). In every replicate plot's inner measurement core, four chamber bases (fabricated from a 250 mm PN10 PVC pipe and each with an area = 0.044 $m^2$, and volume = about 12 L) were randomly installed at the soil surface, to a depth of about 0.03 m. Installation of chamber bases was done at the beginning of April 2019, a month prior to the GHG flux measurements, and chamber bases remained permanently in place for the entire measurement period. Litter was not removed from the chambers. However, all the chamber bases were always maintained vegetation free throughout the gas sampling period in order to avoid measuring plant night respiration during chamber closure. On the sampling day, chamber bases were covered with vented polyvinyl hoods fitted with sampling ports. A pooled gas sample was then obtained every 3, 13, 23, and 33 minutes using an airtight Luer Lock syringe following the pooling approach described in detail by Arias-Navarro et al. (2013). The 33 minute maximum chamber closure period used in this study was well under the threshold recommended by Pavelka et al. (2018)), but, comparable to other tropical GHG flux studies (e.g. Corre et al. (2010), Koehler et al. (2009a), Matson et al. (2017)). To check if the pooling worked correctly, both the pooled and un-pooled (an average of four individual chamber measurements) samples were taken for the month of February 2020 for analysis. Both methods produced very comparable results (Fig. A1). Soil GHG fluxes were always

measured between 9 am and 4 pm throughout the entire study period, while for each measurement day, the sequence of plots to be measured was randomly chosen. Together with the very low diurnal variability of air (0.6 ± 0.04 °C; mean ± SE) and soil (0.2 ± 0.03 °C; mean ± SE) temperatures at this tropical forest site, time of measurement of individual gas chambers should, if at all, only have a minimal effect on the measured gas fluxes. All collected gas samples were stored in Labco Exetainers (Labco Limited, Lempeter, UK) with screw-on plastic caps fitted with Labco Grey Chlorobutyl Septum because these exetainers have been demonstrated to remain airtight for periods spanning up to six months (Hassler et al., 2015). Additionally, all the plastic caps were screwed on to the exetainers by hand and "quarter turned" prior to sampling to ensure that they were airtight (Pavelka et al., 2018). All the gas-filled exetainers were shipped to the Department of Environmental Systems Science, ETH Zürich, Switzerland for analysis using a gas chromatograph (GC; Scion 456-GC Bruker, Germany) within a period of four months from sampling. The GC was equipped with an electron capture detector ($N_2O$), flame ionization detector ($CH_4$), thermal conductivity detector ($CO_2$), and auto-sampler. GC concentrations of the individual gas species of interest ($CO_2$, $CH_4$ and $N_2O$) were then calculated by comparing the peak areas of the measured samples to the respective peak areas of a suite of standard gas samples. Next, flux rates of individual gases at the soil-atmospheric interface were calculated based on either linear increase or decrease in gas concentrations during chamber closure following Eq. 1 in Butterbach-Bahl et al. (2011).

$$GHG_{flux} = \frac{V_{ch} * GHG_m * S * 10^6 * 60}{A_{ch} * GHG_v * 10^9} \qquad (1)$$

where $GHG_{flux}$ is given as a positive flux to the atmosphere or a negative flux into the soil [µg m$^{-2}$ h$^{-1}$], $V_{ch}$ is the chamber volume [m$^3$], $GHG_m$ is the molar mass of the different gases [g mol$^{-1}$], S is the slope of a linear regression calculated based on the increase or decrease in gas concentrations during chamber closure [ppm min$^{-1}$], $A_{ch}$ is the chamber ground area [m$^2$], $GHG_v$ is the molar volume of the different gases [m$^3$ mol$^{-1}$]. Note that the constants $10^6$, $10^9$, and 60 were used to convert grams into micrograms, parts per million into cubic meters, and minutes into hours. $GHG_v$ was adjusted to air temperature and pressure in the field using ideal gas law following Eq. 2:

$$GHG_v = 0.02241 * \frac{273.15 + T_f}{273.15} * \frac{P_f}{P_s} \qquad (2)$$

where $T_f$ is the air temperature [°C] and $P_f$ is the pressure [Pa] at the field site, while $P_s$ is the pressure at sea level [Pa]. As a quality check, the linearity of $CO_2$ increase during chamber closure was inspected by comparing the $CO_2$ concentrations (of each chamber measurement) with time since chamber closure, and thereafter, determined the goodness of fit for the linear regression model (the $R^2$). The $R^2$ for all the measurements was 0.992 ± 0.001 (mean ± SE). Additionally, the measured gas concentrations from the GC were checked against the standards and the GC's minimum detection limit to ensure that the changes in gas concentrations during chamber closure were well above its minimum detection limit.

In parallel to gas flux measurements, soil environmental controls particularly soil temperature, volumetric water content, and soil mineral nitrogen (ammonium ($NH_4^+$) and nitrate ($NO_3^-$)) were measured. Soil temperature and volumetric water content were determined at 0.05 m soil depth adjacent to each of the four-installed chamber bases per replicate plot. A digital thermometer (Greisinger GMH 3230, Germany) fitted with an insertion probe and a calibrated ML3 ThetaProbe soil moisture sensor (AT Delta-T Devices Limited, United Kingdom) were used to determine soil temperature and soil volumetric water content, respectively. Soil mineral nitrogen was determined by

obtaining a soil sample in a Kopercky ring at 0.05 m depth (from the soil surface) and 1 m distance from each of the installed chamber per replicate. The obtained soil samples (from each replicate plot) were pooled together and

thoroughly mixed. Next, 100 and 150 g of the pooled soil samples were extracted with 100 and 600 mL $CaCl_2$ solution for determination of $NO_3^-$ and $NH_4^+$ concentrations respectively using the RQflex® 10 reflectometer. RQflex® 10 reflectometer is part of the reflectoquant system comprising of a reflectometer, batch-specific barcode and test strips. The test strips used in this study had a 3 - 90 and 0.2 - 7 mg $L^{-1}$ detection range for nitrates ($NO_3$-N) and ammonium ($NH_4$-N), respectively.

To understand the contribution of autotrophic (root) and heterotrophic (microbial) sources to total soil respiration, a trenching treatment was done in all the plots following the protocol of Wang and Yang (2007). Prior to trenching, root biomass distribution with depth was determined in order to establish where most roots were located. Root biomass estimation involved digging three profile pits measuring 1 m (L) x 1 m (W) x 1.1 m (D) at the forest site. In every pit, ten soil monoliths (each measuring: 20 cm (L) x 20 cm (W)) were carefully cut out (using a spade and hoe) following

a 10 cm depth interval from the surface down to 1 m. The soil monoliths were thoroughly washed to isolate the roots from the bulk soil. The root samples were oven dried at 60 °C for 48 hours and weighed to determine the root biomass per depth increment. The root biomass for each depth interval was calculated as the mean of the root biomass from the three pits for that interval. It was established that over 90 % of the roots were within the top 0.6 m of the soil profile. Therefore, a circular trench (about 0.60 m in diameter) was dug to a depth of about 0.6 m at the center of all the plots,

thereby creating a soil mass free of roots. All the trenches were lined with a heavy-duty plastic sheet to prevent roots from growing back into the trenched soil mass. The trenched soil mass and the proximally neighboring un-trenched (reference) zone (about 1 m apart) were respectively installed with a chamber base. Both the trenched and reference chamber bases had a design (area = 0.044 $m^2$, and volume = about 12 L) identical to the one used in the NME soil GHG flux study. The installed chamber bases were left standing for six months, before the first measurements began

in November 2019. This ensured that a large proportion of the cut roots in the trenched soil mass decomposed before the start of the $CO_2$ measurements. $CO_2$ measurements were conducted monthly for a period of four (4) months (starting in November 2019 and ending in February 2020). The selected measurement time window represented the transition between the wet season and the long dry season, allowing us to capture how soil moisture constrained the different soil $CO_2$ efflux sources. After the completion of flux measurements, root coring was done to a depth of 0.30 m at two

locations directly adjacent to both the trenched and un-trenched chambers, in order to determine if the trenching approach was effective in reducing the amount of living root biomass in the trenched zone. It was established that there was a 73 % and 63 % reduction in fine root biomass, and coarse root biomass, respectively, in the trenched zone in comparison to the reference zone. Heterotrophic (microbial) respiration was equal to the $CO_2$ effluxes from the trenched chamber while autotrophic (root) respiration was the difference between $CO_2$ effluxes from the reference and trenched

chambers.

### 2.5    Statistical Analysis

Prior to statistical analysis, transitory $N_2O$ fluxes from N addition plots (N, and N + P) were detrended to compensate for absence of frequent measurements immediately after fertilization coming from sampling GHGs monthly.

Detrending involved using a log-normal fit between the measured $N_2O$ fluxes and time since fertilization (until day

42), and this explained 43 % of the observed variability in the $N_2O$ data during the transitory phase ($p < 0.05$). Additionally, GHG flux and soil environmental control data were aggregated based on seasons (wet and dry) and phases (transitory; 0-28 days from the date of fertilization, and background; more than 28 days after fertilization). Furthermore, despite monitoring soil $NO_3^-$ and $NH_4^+$ contents on a monthly basis throughout the measurement period, only the soil $NO_3^-$ data set was used in the analysis because soil $NH_4^+$ was mostly low and often below the detection limit of the reflectometer at majority of the sampling time points.

Data was checked for normality and homogeneity of variance (homoscedasticity) across treatment groups, seasons, and phases before implementing parametric tests (i.e. linear mixed effects model (LMEMs), and one-way analysis of variance (ANOVA)). Normality of the respective data was inspected by use of diagnostic plots (histograms and quantile-quantile plots), and the Shapiro-Wilk normality test, while heteroscedasticity was determined with the Levene test and by inspecting residual plots of fitted values. In case of heteroscedasticity and non-normal distribution of the data, either a logarithmic or a Tukey transformation was applied on the dataset. However, if normality of the data and homogeneity of variance were not restored by the transformations, an equivalent non-parametric statistical test was selected. The Spearman's correlation coefficient test was used to check the relationship between the measured background soil GHG fluxes and soil environmental controls.

To determine differences in mean soil GHG fluxes between treatments, one way ANOVA test was used with GHG species and treatments included in the model as response and predictor variables, respectively. In order to determine the effect of the added nutrients on soil GHG fluxes ($CO_2$, $CH_4$, and $N_2O$), soil $CO_2$ sources (heterotrophic and autotrophic), and soil environmental controls (water filled pore space, soil temperature, and nitrates), LMEMs were employed. LMEMs effectively deal with temporal pseudo-replication (coming from repeated measurements) hence safeguard against inflation of the degrees of freedom, which would significantly compromise the power of the statistical test. Added nutrients (treatments), seasons (wet and dry), $CO_2$ sources (autotrophic and heterotrophic) and phases (transitory and background) were included in the LMEMs as fixed effects while sampling days and replicate plots were included as random effects. Some of the LMEMs were extended to either include a variance function (to account for variation in the response variable per level of the fixed effect), or a first order temporal auto regressive process (to control for correlation between closely spaced measurements in time) or both. The extensions were included in the LMEMs on the premise that they improved the relative goodness of model fit based on Akaike Information Criteria (AIC).

All the statistical data analyses were performed using R 3.6.3 (R Development Core Team, 2019). Specifically, *nlme' and 'car'* packages were employed to run LMEMs and one-way ANOVA tests, respectively. Throughout the paper, statistical significance in all the tests was inferred if $p \leq 0.05$. Annual soil GHG fluxes were estimated through a trapezoidal interpolation on the measured monthly soil GHG fluxes.

## 3    Results

### 3.1    Soil physico-chemical characteristics, water filled pore space, soil temperature and nitrates

Soil characteristics did not significantly differ across plots; hence, the parameters presented in Table 1 represent the soil physico-chemical characteristic for the NME site.

**Table 1. Soil physico-chemical properties in three depths and vegetation characteristics of the study site located in Budongo forest, northwestern Uganda.**

| Soil physico-chemical properties | Soil depth (m) | | |
|---|---|---|---|
| | 0 - 0.10 | 0.10 - 0.30 | 0.30 - 0.50 |
| Soil bulk density (g cm$^{-3}$) | $1.2 \pm 0.2$ | $1.5 \pm 0.2$ | $1.3 \pm 0.2$ |
| Soil pH (1:2.5) | $6.4 \pm 0.2$ | $6.2 \pm 0.2$ | $6.0 \pm 0.2$ |
| Soil total carbon (C) (kg C m$^{-2}$) | $4.1 \pm 0.0$ | $3.1 \pm 0.0$ | $1.8 \pm 0.0$ |
| Soil total nitrogen (N) (g N m$^{-2}$) | $423 \pm 1.0$ | $387 \pm 0.2$ | $249 \pm 0.6$ |
| Soil C/N ratio | $9.5 \pm 0.3$ | $8.0 \pm 0.3$ | $7.2 \pm 0.3$ |
| Sand (%) | $55 \pm 2$ | $55 \pm 2$ | $49 \pm 1$ |
| Silt (%) | $27 \pm 2$ | $21 \pm 1$ | $14 \pm 1$ |
| Clay (%) | $18 \pm 1$ | $23 \pm 1$ | $38 \pm 1$ |
| ECEC (mmol$_c$ kg$^{-1}$) | $149 \pm 8$ | $76 \pm 4$ | $62 \pm 4$ |
| Exchangeable aluminum (g Al m$^{-2}$) | $0.10 \pm 0.06$ | $0.11 \pm 0.15$ | $0.14 \pm 0.20$ |
| Exchangeable calcium (g Ca m$^{-2}$) | $75.6 \pm 4.10$ | $39.0 \pm 8.51$ | $34.7 \pm 8.59$ |
| Exchangeable magnesium (g Mg m$^{-2}$) | $17.0 \pm 0.90$ | $12.3 \pm 2.7$ | $11.7 \pm 1.0$ |
| Bray II extractable phosphorus (g P m$^{-2}$) | $1.80 \pm 0.20$ | $1.01 \pm 0.14$ | $0.838 \pm 0.159$ |
| Base saturation (%) | $99 \pm 1$ | $97 \pm 1$ | $98 \pm 1$ |
| Plant-available phosphorus (g P m$^{-2}$) | $1.7 \pm 0.0$ | - | - |
| Plant-available molybdenum (mg Mo m$^{-2}$) | $14 \pm 5.0$ | - | - |
| Vegetation characteristics ($\geq$ 10 cm DBH) | | | |
| Forest type | Moist semi-deciduous tropical forest | | |
| Most abundant tree species | Funtumia elastica, Celtis mildbraedii, Cynometra alexandri, Celtis zenkeri | | |
| Stand height (m) | $18.7 \pm 0.1$ | | |
| Mean basal area (m$^2$ ha$^{-1}$) | $34.0 \pm 1.0$ | | |
| Tree density (trees ha$^{-1}$) | $621 \pm 13$ | | |
| N fixing trees at the site (trees ha$^{-1}$) | $\sim 42$ | | |

Notes: DBH is diameter at breast height. ECEC is effective cation exchange capacity.

300

The soils have a high bulk density (specifically 10 - 30 cm), slightly acidic pH, sandy texture, relatively high effective cation exchange capacity (ECEC), high base saturation (dominated by Ca and Mg), low in plant available phosphorus, and a low C/N (Table 1). Water filled pore space (WFPS) was significantly higher in the wet season (March to December; $55 \pm 1.0$ %) compared to the dry season (January to February; $43 \pm 1.7$ %) (Fig. 1a, Fig. 2a, $p < 0.01$).

305 WFPS was higher in N and N + P addition plots compared the control plots both in the dry (N; $p = 0.02$, N + P; $p = 0.04$) and wet (N; $p = 0.02$, N + P; $p = 0.05$) seasons (Fig. 1a). Soil temperature varied minimally (0.6 °C) across treatments and seasons ranging between 20.1 and 21.4 °C in the dry season, and between 19.7 and 22.9 °C in the wet season. Soil nitrate contents measured across all treatment plots were significantly larger in the dry season compared to the wet season (Fig. 1c, $p < 0.01$). Soil nitrate content from the N ($p = 0.01$) and N + P ($p = 0.02$) addition plots was

310 significantly higher than the control plots in the wet season (Fig.1c), but no significant difference was detected between the nutrient addition treatments and the control in the dry season (Fig. 1c). Strong nitrate peaks were observed in N and N + P addition plots in September 2019 and June 2020 shortly after fertilization (Fig. 2c).

**Table 2.** Mean (± SE, n = 4) soil GHG fluxes ($CO_2$, $CH_4$, $N_2O$) as well as annual soil GHG fluxes measured between May 2019 and June 2020 from control (Ctrl), nitrogen (N), phosphorus (P), and N + P plots of the nutrient manipulation experiment.

| Treatment [a] | $CO_2$ fluxes (mg C m$^{-2}$ h$^{-1}$) | Annual $CO_2$ fluxes† (Mg C ha$^{-1}$ yr$^{-1}$) | $CH_4$ fluxes (µg C m$^{-2}$ h$^{-1}$) | Annual $CH_4$ fluxes† (kg C ha$^{-1}$ yr$^{-1}$) | $N_2O$ fluxes (µg N m$^{-2}$ h$^{-1}$) | Annual $N_2O$ fluxes† (kg N ha$^{-1}$ yr$^{-1}$) |
|---|---|---|---|---|---|---|
| Ctrl | 164 ± 5.3[a] | 14.5 ± 0.6[a] | -30.5 ± 4.9[a] | -2.7 ± 0.4[a] | 20.5 ± 3.2[a] | 1.8 ± 0.3[a] |
| N | 186 ± 6.5[a] | 16.4 ± 0.9[a] | -39.7 ± 4.4[a] | -3.4 ± 0.4[a] | 50.2 ± 11[b] | 4.8 ± 1.5[b] |
| P | 186 ± 5.3[a] | 16.4 ± 1.0[a] | -56.2 ± 3.8[b] | -4.7 ± 0.7[b] | 21.8 ± 2.4[a] | 1.9 ± 0.3[a] |
| N + P | 197 ± 5.4[b] | 17.3 ± 0.8[b] | -39.3 ± 6.3[a] | -3.3 ± 0.7[a] | 53.8 ± 10[b] | 4.6 ± 0.4[b] |

Notes: [a]Means followed by different lower-case letters indicate significant differences among treatments (One way analysis of variance, $p \leq 0.05$); †Annual soil $CO_2$ fluxes, $CH_4$ fluxes, and $N_2O$ fluxes were approximated by applying the trapezoid rule on time intervals between measured flux rates. The mean and annual soil GHG fluxes included both transitory and background flux measurements. Note: Transitory $N_2O$ fluxes (measured within 28 days from fertilization) from N addition plots (N, N + P) were detrended to compensate for absence of frequent measurements immediately after fertilization coming from sampling GHGs monthly.

### 3.2 Soil $CO_2$ fluxes

Soil $CO_2$ fluxes varied between 60 and 330 mg C m$^{-2}$ h$^{-1}$ during the measurement period across all treatments. However, the highest $CO_2$ fluxes were measured in December at the interface between wet and dry season (Fig. 3a). Fertilization resulted in an immediate increase in $CO_2$ fluxes across all nutrient addition plots (N; 15 %, P; 14 %, N + P; 24 %) in the transitory phase. However, this increase was only significant in the N + P plots (Fig. 4a, $p = 0.01$). There was no significant effect of fertilization on background $CO_2$ fluxes between nutrient addition treatments and the control plots (Fig. 4d).

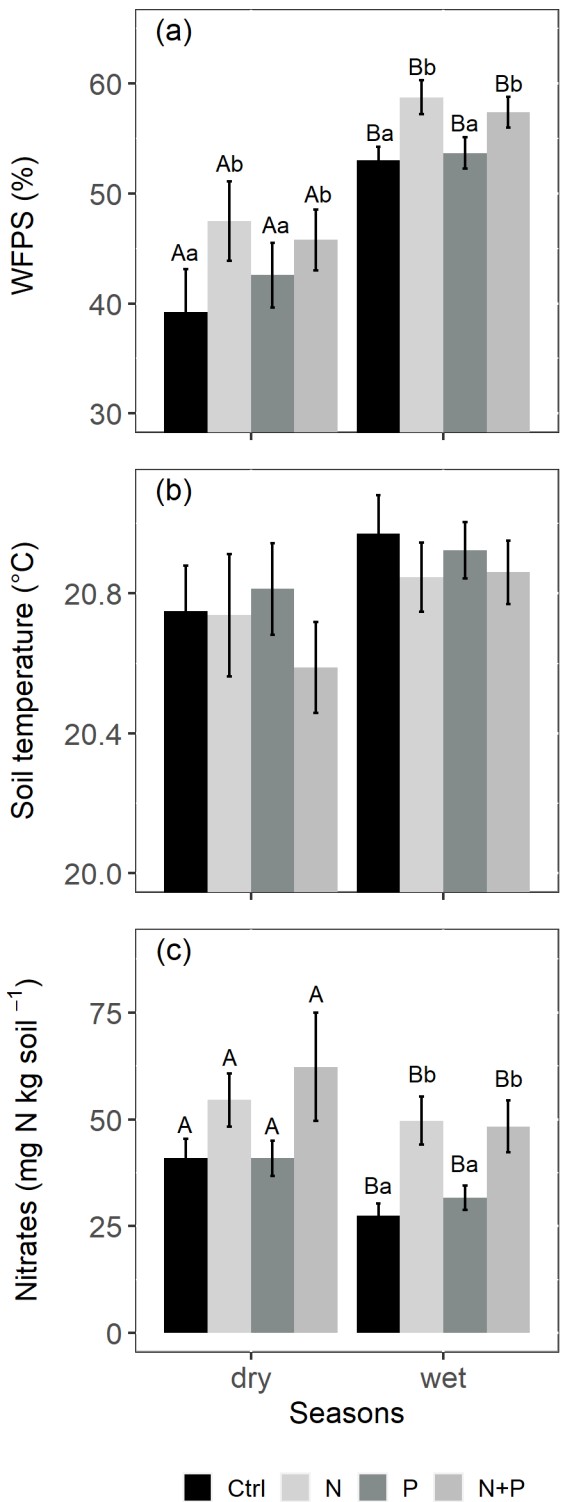

**Figure 1. Mean (± SE, n = 4) water filled pore space (WFPS) (a), soil temperature (b), and nitrates (c) in the top 0.05 m of the control (Ctrl), nitrogen (N), phosphorus (P), and N + P plots of the nutrient manipulation experiment measured during the dry (January and February; monthly precipitation < 100 mm) and wet (March to December; monthly precipitation >**

Similarly, no significant differences in the background $CO_2$ fluxes were detected between seasons despite measuring marginally lower background $CO_2$ fluxes in the wet season compared to the dry season (Fig. 4d). Additionally, no significant differences were detected between transitory and background $CO_2$ fluxes (Fig. 4a, d). Heterotrophic

(microbial) $CO_2$ effluxes were significantly higher than the autotrophic (root) $CO_2$ effluxes (Fig. 5, _p < 0.01_) across all treatment plots with microbes contributing about three times more to the total soil $CO_2$ effluxes compared to roots (Fig. 5, _p < 0.01_). Neither heterotrophic nor autotrophic respiration significantly differed between treatments (Fig. 5). Overall, there was a relatively low variability in annual $CO_2$ fluxes across treatments (CV = 14.8 ± 2.2 %). The Spearman's correlation coefficient indicated that background soil $CO_2$ fluxes did not correlate to any of the measured

soil environmental controls (WFPS, soil temperature, and nitrates) across all treatment plots (Fig. 6a, b, c).

### 3.3 Soil CH₄ fluxes

Across all treatments, phases (transitory and background) and seasons, soil $CH_4$ fluxes varied between an uptake of -278 mg C m$^{-2}$ h$^{-1}$ and a release of 77 mg C m$^{-2}$ h$^{-1}$. In the transitory phase, $CH_4$ consumption increased slightly but not

significantly in the N (2 %) and N + P (6 %) plots. A larger but still not significant (marginal) increase was found in the case of P plots (54 %; _p = 0.05_) (Fig. 4b). Beyond 28 days from fertilization, no significant difference in background soil $CH_4$ fluxes between treatments was detected in the dry season (Fig. 4e). However, a significantly higher background soil $CH_4$ consumption was measured in P plots in the wet season (Fig. 4e, _p = 0.01_). Soil $CH_4$ consumption in the dry season was on average 1.5 times larger than the wet season across all treatments (Fig. 4e, _p = 0.01_). Soil

$CH_4$ uptake across all treatment plots measured during the transitory phase (-39.0 ± 3.7 mg C m$^{-2}$ h$^{-1}$) did not significantly differ from the $CH_4$ uptake in the background phase (-42.8 ± 3.4 mg C m$^{-2}$ h$^{-1}$) (Fig. 4b, e). Annual $CH_4$ uptake ranged between -2.7 and -4.7 kg C ha$^{-1}$ yr$^{-1}$, with soils in all the treatment plots acting as net sinks for $CH_4$ (Table 2). The Spearman's correlation coefficient test indicated that background $CH_4$ fluxes were strongly and positively correlated to WFPS (Fig. 6d) while soil temperature (Fig. 6e) and nitrates (Fig. 6f) were also significant but

negatively correlated.

### 3.4 Soil N₂O fluxes

Soil $N_2O$ fluxes across treatments, phases (transitory and background), and seasons varied between an uptake of -18 µg N m$^{-2}$ h$^{-1}$ and a release of 499 µg N m$^{-2}$ h$^{-1}$. A strong increase in $N_2O$ effluxes was measured immediately after

fertilization (September and December 2019, April and June 2020) in all N addition plots with increases of 445 % in N plots (_p < 0.01_) and 455 % in the N + P plots (_p < 0.01_) compared to the control plots in the transitory phase (Fig. 4c). The soil $N_2O$ peaks in September 2019 and June 2020 (Fig. 3c) coincided with the peaking in soil nitrate concentrations (Fig. 2c). Background soil $N_2O$ fluxes did not differ significantly between nutrient addition plots and the control plots both in the dry and wet seasons (Fig. 4f). Annual $N_2O$ fluxes ranged between 1.8 and 4.8 kg N ha$^{-1}$

yr$^{-1}$, with soils in all the treatment plots acting as net sources of $N_2O$ (Table 2). The Spearman's correlation coefficient

indicated that background soil $N_2O$ fluxes were strongly and positively correlated to WFPS (Fig. 6g) in all treatment plots. Majority of the background soil $N_2O$ fluxes higher than 15 µg N m$^{-2}$ h$^{-1}$ (constituting 74 % of the averages background soil $N_2O$ fluxes) corresponded to WFPS greater than 49 % (wetter conditions) (Fig. 6g). Background soil $N_2O$ fluxes negatively correlated to soil temperature (Fig. 6h) and nitrates (Fig. 6i) in all treatment plots.

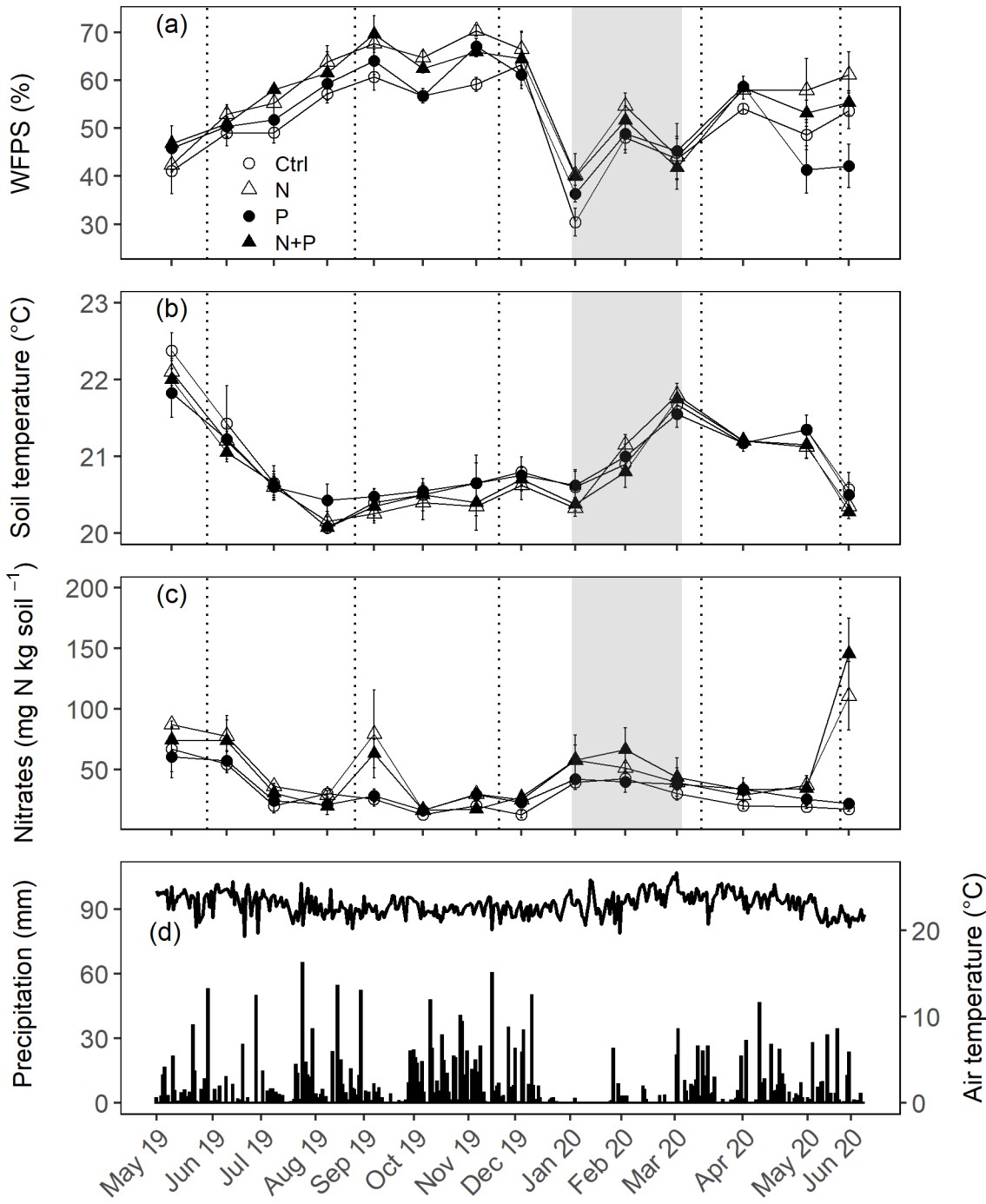


**Figure 2. Mean (± SE, n = 4) water filled pore space (a), soil temperature (b), and nitrates (c) in the top 0.05 m measured monthly (May 2019 to June 2020) from control (Ctrl), nitrogen (N), phosphorus (P), and N + P plots of the nutrient**

manipulation experiment. Vertical lines indicate the timing of each split dose of N (31.3 kg N ha$^{-1}$), P (12.5 kg P ha$^{-1}$) and N (31.3 kg N ha$^{-1}$) + P (12.5 kg P ha$^{-1}$) fertilization every three months. The gray shaded rectangle (in a, b, and c) marks the beginning and end of the dry season (January and February; monthly precipitation < 100 mm), while (d) gives the daily precipitation (bars) and air temperature (line) between May 2019 and June 2020. Climatic data was obtained from a weather station installed at Budongo Conservation Field Station, 2 km from the location of the nutrient manipulation experiment in Budongo forest, northwestern Uganda.

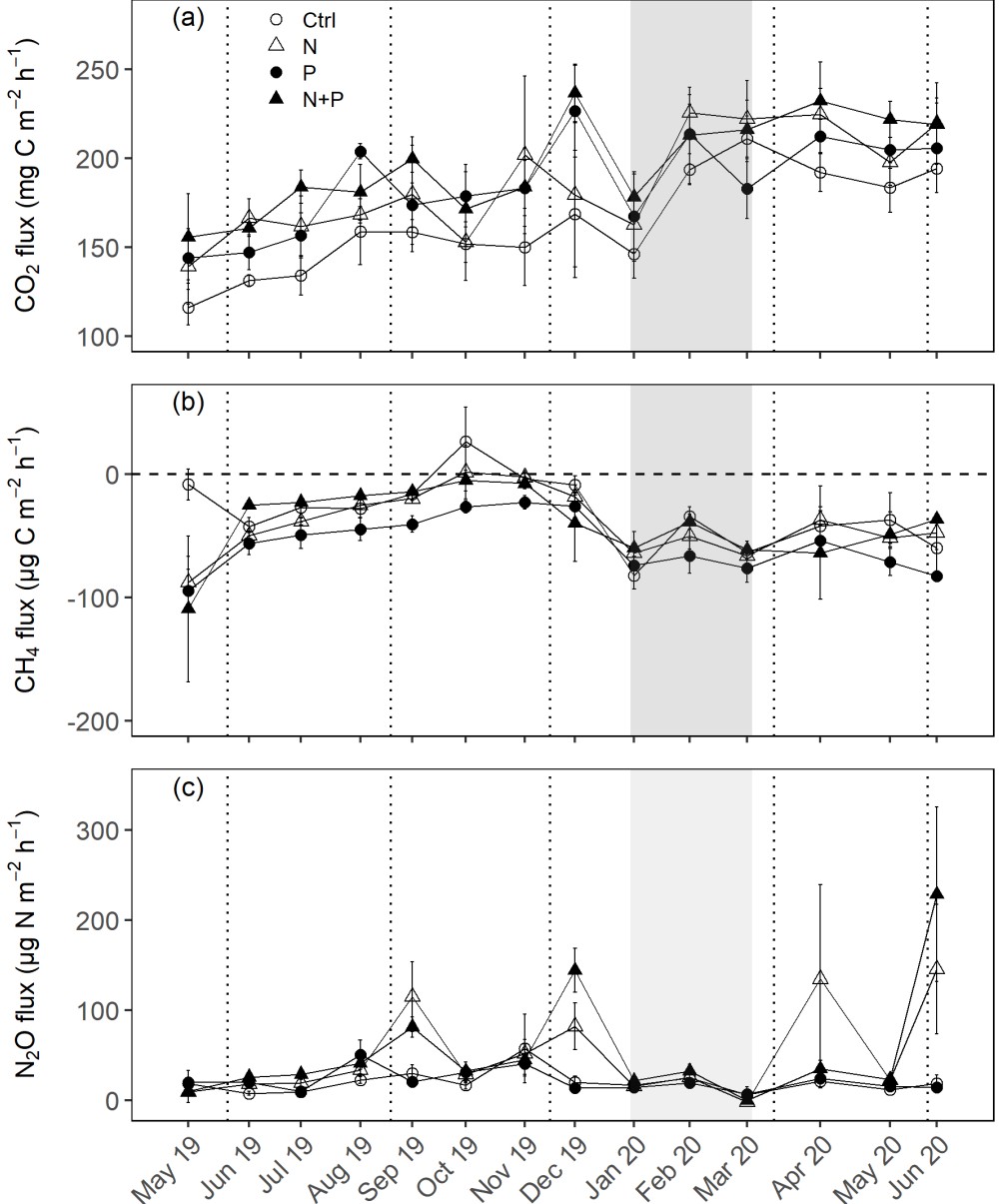

Figure 3. Mean (± SE, n = 4) soil $CO_2$ fluxes (a), $CH_4$ fluxes (b), and $N_2O$ fluxes (c) measured monthly (between May 2019 and June 2020) from control (Ctrl), nitrogen (N), phosphorus (P), and N + P plots of the nutrient manipulation experiment. Vertical lines indicate the timing of each split dose of N (31.3 kg N ha$^{-1}$), P (12.5 kg P ha$^{-1}$) and N (31.3 kg N ha$^{-1}$) + P (12.5 kg P ha$^{-1}$) fertilization every three months. The gray shaded rectangle marks the beginning and end of the dry season (January and February; monthly precipitation < 100 mm). Note: Transitory $N_2O$ fluxes (measured within 28 days from

 **fertilization) from N addition plots (N, N + P) were detrended to compensate for absence of frequent measurements immediately after fertilization coming from sampling GHGs monthly.**

## 4    Discussion

### 4.1    Effect of N and P addition and soil environmental controls on soil $CO_2$ fluxes

The annual soil $CO_2$ effluxes from control plots (Table 2) were lower than those measured from tropical forests in Thailand (Hashimoto et al., 2004) and Hawaii (Townsend et al., 1995); within range to those from the Democratic Republic of Congo (Baumgartner et al., 2020), Panama (Koehler et al., 2009a; Pendall et al., 2010), Brazil (Sousa Neto et al., 2011), and Cameroon (Verchot et al., 2020); and higher than those reported from Kenya (Wanyama et al., 2019), and Indonesia (van Straaten et al., 2011). The differences in soil $CO_2$ fluxes between the control plots in this study and studies done in other tropical forest sites may be due to differences in soil environmental characteristics e.g. soil C quality and quantity, soil temperature, and moisture availability at the respective sites (Nottingham et al., 2015).

The alleviation of nutrient limitations on soil biological activity (in microbial communities and in root respiration) through fertilizer addition was particularly reflected by the significant increase in transitory $CO_2$ effluxes following addition of both N and P together (Fig. 4a). The transitory phase (< 28 days from fertilization) is the period where addition of nutrients (N, P, N + P) is expected to result in a large pulse of microbial activities. However, the fact that the increase in soil $CO_2$ effluxes was significant only in plots where N and P were added simultaneously (N + P), suggests a possible co-limitation between N and P on soil biological activity (Bréchet et al., 2019). These results seemingly align with the proposed multiple element limitation concept, which suggests a strong response in microbial mediated processes upon supply of limiting nutrients (Fanin et al., 2015). Furthermore, the results likely indicate that some soil respiration sources may respond positively to N addition (Yan et al., 2017), while others may respond positively to P addition (Ma et al., 2020), yielding an overall additive response when added together. An increase in soil $CO_2$ effluxes following addition of N and P simultaneously together has also been reported in studies like Bréchet et al. (2019), and Soong et al. (2018) from Panamanian tropical forests.

In contrast, the lack of significant treatment effects on background soil $CO_2$ efflux (Fig. 4a, d) and its different components (heterotrophic and autotrophic; Fig. 5) may suggest that numerous counteracting processes could be happening at the same time, hence masking treatment effects. Some studies have for instance demonstrated that addition of N subdues exoenzymes (Li et al., 2018), decreases microbial biomass (Burton et al., 2004; Hicks et al., 2019), increases net primary productivity (Adamek et al., 2009), reduces fine root biomass (Cusack et al., 2011), while other studies have reported that P addition increases soil organic matter decompsition in tropical forest ecosytems (Cleveland and Townsend, 2006). The possibility of counteracting processes at the experimental site is further exemplified by the lack of a relationship between all the measured soil environmental controls (soil temperature, nitrates and soil moisture) and background $CO_2$ effluxes (Fig 6a, b, c). Although these results are consistent with the findings by Baumgartner et al. (2020) in the Congo basin, they contrast several GHG studies located in tropical forests that have reported a strong correlation between $CO_2$ effluxes and soil moisture (Matson et al., 2017; van Straaten et al., 2011). For this experiment site, it could be that the minimal temporal fluctuation in soil temperature (Fig. 1b),

together with the fact that water filled pore space was mostly > 40 % (Fig. 1a) during the sampling campaign dampened the effect of soil temperature and moisture on soil $CO_2$ fluxes.

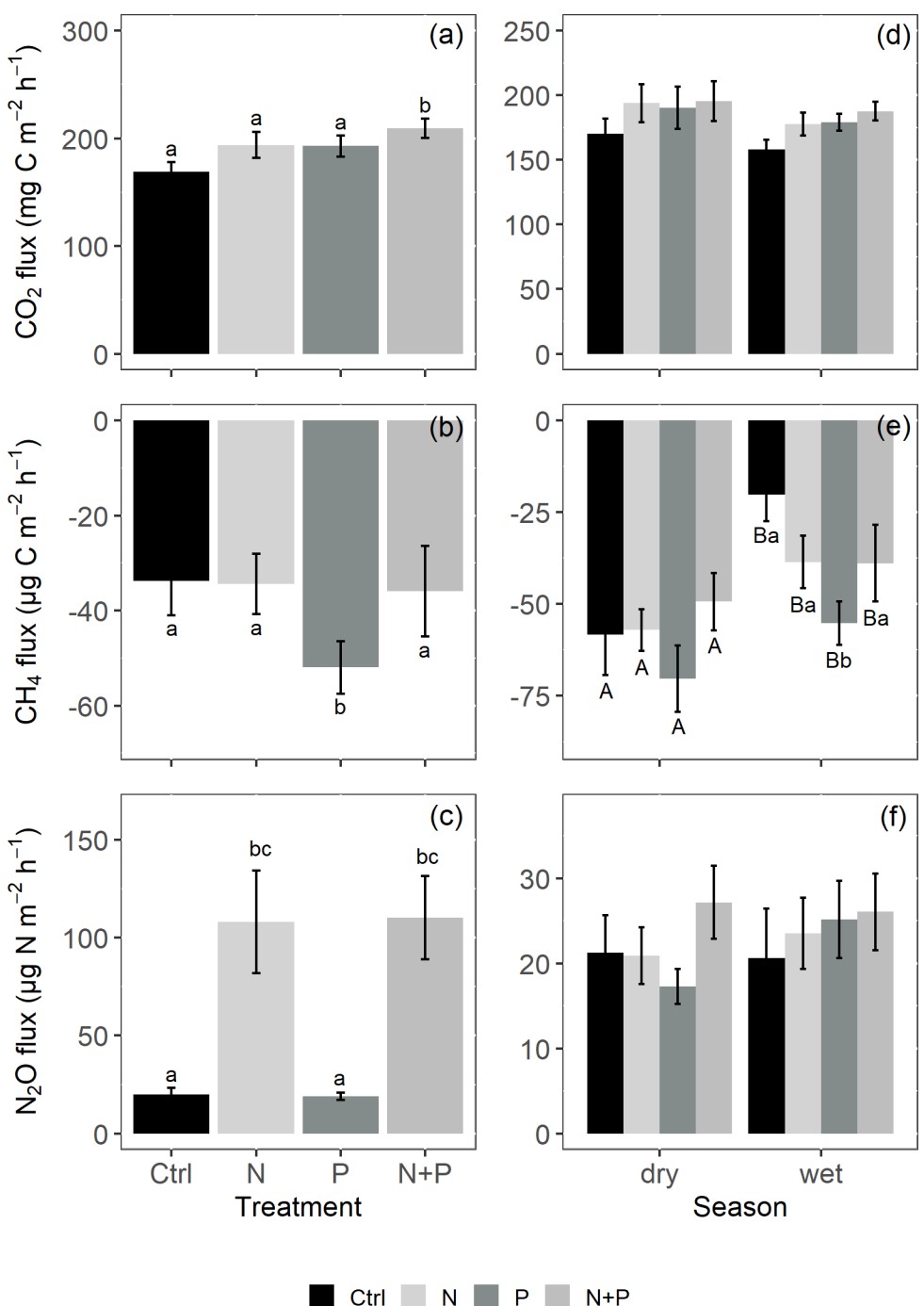

 **Figure 4. Mean (± SE, n = 4) soil $CO_2$ fluxes (a, d), $CH_4$ fluxes (b, e), and $N_2O$ fluxes (c, f) from the control (Ctrl), nitrogen (N), phosphorus (P), and N + P plots of the nutrient manipulation experiment. Column 1 (a, b, and c) includes only fluxes measured during the transitory phase (0 to 28 days after fertilization; and all the transitory fluxes were in the wet season**

(monthly precipitation >100 mm)). Column 2 (d, e, and f) includes only background level fluxes (fluxes measured more than 28 days after fertilization). Different lower-case letters indicate significant differences between nutrient addition treatments and the control while different upper-case letters indicate significant differences between seasons (linear mixed effects models; *p ≤ 0.05*). Note: Transitory $N_2O$ fluxes (measured within 28 days from fertilization) from N addition plots (N, N + P) were detrended to compensate for absence of frequent measurements immediately after fertilization coming from sampling GHGs monthly.

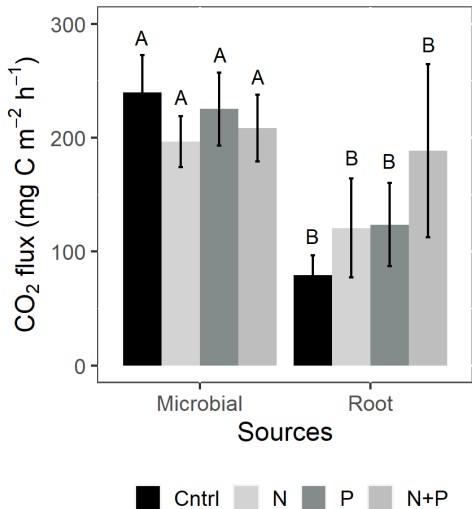

**Figure 5. Mean (± SE, n = 4) soil $CO_2$ flux from the control (Ctrl), nitrogen (N), phosphorus (P), and N + P plots of a trenching treatment separated into microbial and root sources. Different upper-case letters indicate significant differences between microbial and root contribution to total $CO_2$ flux (linear mixed effects models; *p ≤ 0.05*).**

## 4.2 Effect of N and P addition and soil environmental controls on soil $CH_4$ fluxes

The annual soil $CH_4$ fluxes from the control plots (Table 2) were at the upper end of $CH_4$ fluxes measured in lowland tropical forests (Aronson et al., 2019; Veldkamp et al., 2013; Zheng et al., 2016), and at the lower end of those measured in (sub-) montane tropical forest ecosystems (Sousa Neto et al., 2011; Yan et al., 2008). The difference in soil texture and soil moisture regimes between this experimental site and the other study sites might explain why $CH_4$ uptake at the respective sites was different. It is recognized that soil physical properties, particularly texture (Sousa Neto et al., 2011), along with soil moisture content directly control the entry and diffusivity of $CH_4$ from the atmosphere to the oxidative sites in the soil (Veldkamp et al., 2013).

In this experiment, the significantly higher $CH_4$ consumption from the P addition plots compared to the control during both the transitory and background periods (Fig. 4b, e) is attributed to the alleviation of P limitations affecting methanotrophic activity. Similar findings were reported by Zhang et al. (2011), and Yu et al. (2017), but contrasted those of Bréchet et al. (2019) and Zheng et al. (2016). It is worth noting that although all these studies were located in tropical forests, they differed fundamentally in their experimental designs, type and amount of fertilizers applied, and

the frequency of fertilizer application, which could have influenced the reported $CH_4$ uptake rates at the respective

sites.

The lack of a response in background $CH_4$ consumption following N fertilization (Fig. 4e) is likely because there were contrasting ecosystem responses to N addition. On the one hand, the addition of nitrogen significantly increased soil water filled pore space in comparison to the control (Fig. 1a; possibly as a result of reduced fine root biomass (Cusack et al., 2011)), which could have resulted in a decrease in methane uptake. On the other hand, the negative correlation

between nitrates and background $CH_4$ fluxes (Fig. 6f) indicates that increases in soil nitrate content should increase $CH_4$ uptake. Additionally, the lack of a clearer signal in background $CH_4$ uptake may have to do with the high variability in the measured $CH_4$ fluxes (CV = 97 ± 58 %) potentially caused by localized termite activity (Brune, 2014; Nauer et al., 2018).

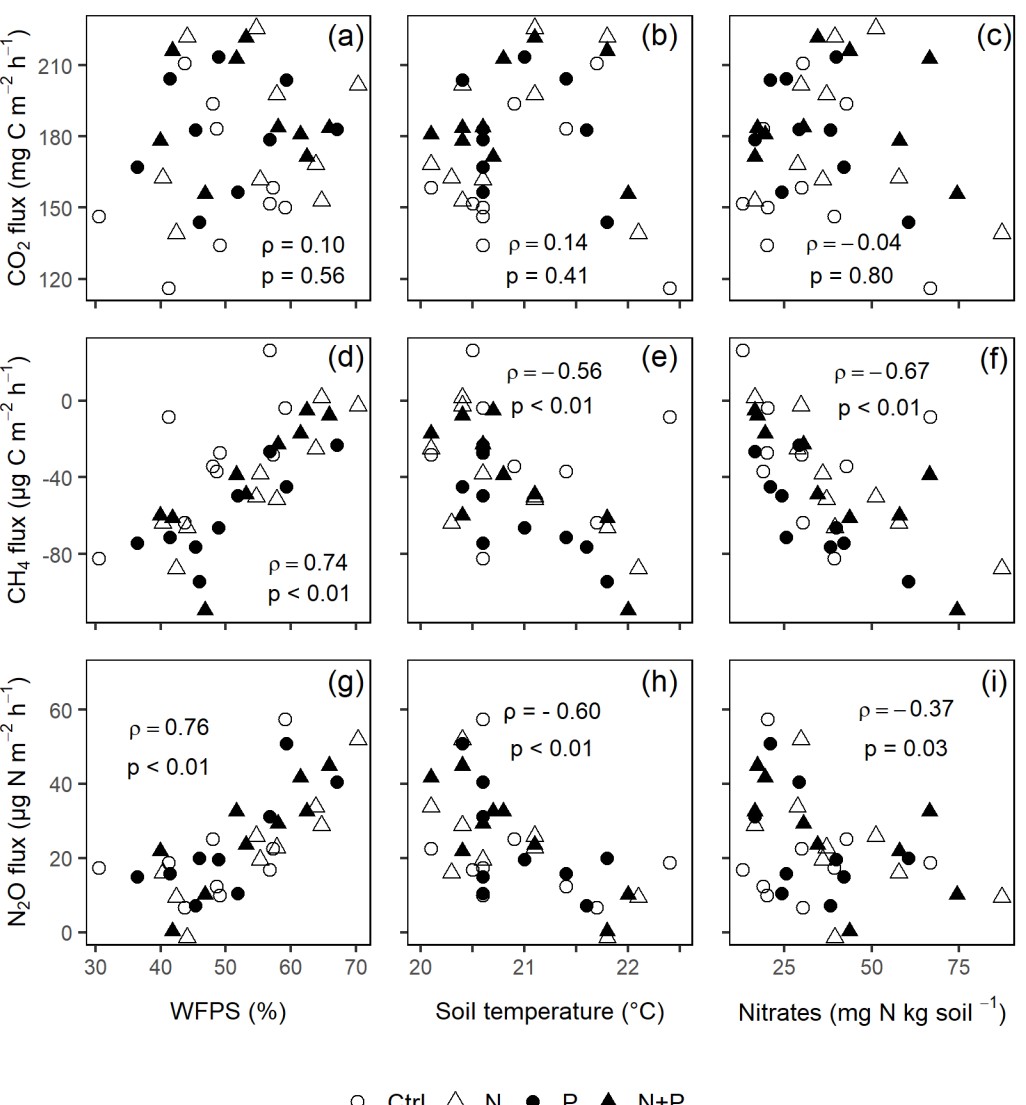


**Figure 6. Spearman's correlation coefficient between mean background $CO_2$ (a-c), $CH_4$ (d-f), and $N_2O$ (g-i) fluxes and WFPS (column 1), soil temperature (column 2) and nitrates (column 3) using monthly measurement means of four replicate**


treatment plots taken between May 2019 and June 2020 ($p \leq 0.05$, n = 16 (four replicate plots in each of the four treatments)) . ρ is the spearman's correlation coefficient.

### 4.3    Effect of N and P addition and soil environmental controls on soil $N_2O$ fluxes

The annual soil $N_2O$ fluxes from the control plots (Table 2) were at the higher end of those measured in (sub-) montane tropical forests (Iddris et al., 2020, Arias-Navarro et al., 2017, Gütlein et al., 2018), and at the lower end of those measured in lowland tropical forest sites (e.g. Koehler et al., 2009b). This may either be due to the differences in soil

N cycling rates (Koehler et al., 2009b) or the differences in spatial abundance of leguminous trees (Xu et al., 2020) at the respective sites.

The immediate flush of $N_2O$ following fertilization (in the transitory phase) both in the N and N + P addition plots (Fig. 3c, Fig. 4c), is due to the increase in soil N concentrations beyond microbial immobilization and plant N needs (Davidson et al., 2000), which is typical of an open or leaky N cycle (Koehler et al., 2009b). Contrary to Kaspari et al.

(2008) and Koehler et al. (2009b), sustained N fertilization did not trigger a significant response in background soil $N_2O$ fluxes from N addition plots (Fig. 4f). This was unexpected, but given the rapid drainage at the site (sandy texture, Table 1), there could have been substantial loss of added N via leaching, which possibly rid the ecosystem of excess nitrates (Lohse and Matson, 2005; Martinson et al., 2013). Notably, sustained P addition did not result in increased background $N_2O$ fluxes (Fig. 4f), which contrasts the findings by Mori et al. (2017) who reported that P availability

opens up the N cycle by stimulating mineralization of soil organic matter releasing excess N that is lost as $N_2O$ emissions. At this study site, it could be that either the amount of P added in the experiment was not sufficient to trigger a response in background soil $N_2O$ fluxes or P is not a limiting nutrient for $N_2O$ fluxes given the relatively high pH of the site (Table 1).

Unexpectedly, nitrates correlated negatively to background $N_2O$ fluxes (Fig. 6i), yet many studies (e.g. Corre et al.,

2014; Zhang et al., 2020) have found that nitrates and $N_2O$ fluxes were positively correlated. The likely explanation for such a relationship is the transformation of $N_2O$ to $N_2$ under wet conditions, which further reduced the amount of nitrates in soil (Matson et al., 2017). Despite the minimal influence of seasonality on background $N_2O$ fluxes (Fig. 4f), a strong positive correlation between background $N_2O$ fluxes and WFPS was observed (Fig 6g), which conforms to the explanation given by the conceptual hole in the pipe (HIP) model. The HIP model places soil aeration status

(approximated by WFPS) second to N availability in controlling soil $N_2O$ fluxes. Soil aeration not only directly controls oxygen entry into the soil but also determines how $N_2O$ is produced (denitrification or nitrification), and transported out of the soil (Davidson et al., 2000). Whereas there seems to be a balance between denitrification and nitrification process at this forest site (given that majority of the measurements corresponded to WFPS of $\leq 60$ %, Fig. 6g), the considerable $N_2O$ fluxes at higher WFPS values ($\geq 60$ %, Fig. 6g) seem to suggest that denitrification is more dominant

than nitrification in producing $N_2O$ in these biomes.

### 4.4    Implications of increasing N and P deposition rates on soil greenhouse gases from tropical forests

While this experiment was established to investigate how nutrient limitations constrain soil GHG fluxes, it also sheds valuable insights on how anthropogenic nutrient inputs (through deposition) may affect future soil GHG fluxes from

African tropical forests and other tropical sites with a similarly strong seasonality, soil, and vegetation characteristics (Table 1). Nutrient depositions are often highest immediately after the onset of the rainy season (Wang et al., 2020), especially due to aerosol deposition following burning activities associated with deforestation during the dry season (Giglio et al., 2006; Roberts et al., 2009). Accordingly, we suspect that the increased N inputs during this short time may yield similar responses to those observed in the transitory period measured at this study site; namely—$N_2O$ flushes when reactive nitrogen enters the soil. Although N additions did not elicit a positive $N_2O$ response during the background period, it is quite likely that our fertilization activities (from year 1 to year 2 of the study) had not gone on for long enough to simulate chronic long-term N additions. A study conducted by Koehler et al. (2009b) in Panama showed that 11 years of chronic N addition significantly increased both transitory and background soil $N_2O$ emissions. In addition, this study shows that future increases in P deposition over tropical forests may significantly increase the $CH_4$ sink capacity of tropical forest soils. Also, it was interesting to observe that the addition of N and P simultaneously resulted in increased $CO_2$ effluxes immediately after fertilization likely suggesting a co-limitation of N and P on soil respiration. This means that future increases in deposition of N and P rich ashes (from biomass burning), might result in significant soil $CO_2$ emissions from these biomes, while it is unclear if this is compensated via an increase in photosynthetic $CO_2$ uptake as indicated by Cernusak et al. (2013). In this context, it is important to note that it has been demonstrated by Barkley et al. (2019) that P derived from biomass burning aerosols is more soluble than the P from dust aerosols, hence, the former would have an immediate impact on ecosystem processes.

**5. Conclusion**

Nutrient manipulation studies premised in tropical forests are crucial to understand how these under-studied yet very important sinks and sources of soil GHGs subtly respond to changes in soil macro nutrient availability. N fertilization (N and N + P) significantly increased $N_2O$ fluxes immediately after fertilization (transitory phase), but had no significant effect on background $N_2O$ fluxes, which might occur if the system would gain N over longer time spans. Against our expectations, neither background $CO_2$ effluxes nor $CH_4$ uptake decreased following addition of N, indicating neither a negative effect of a potential surplus of soil N on root and microbial respiration nor a negative effect on methanotrophs. $CO_2$ effluxes even showed a significant increase during the transitory phase following N and N + P fertilization. However, this effect was only significant for N + P addition, indicating some N and P co-limitation. An increase in $CH_4$ uptake was found both shortly and after sustained P fertilization; supporting our second hypothesis, which suggested that lifting the P limitation on soil methanotrophs, would significantly increase $CH_4$ consumption. Surprisingly, both transitory and background $N_2O$ and $CO_2$ fluxes (including its different components) were not significantly affected by P fertilization. Overall, the results from this first nutrient manipulation GHG study from a wet African tropical forest site, in general, indicate our limited knowledge about the counteracting interactions between N and P inputs and GHG fluxes from different tropical forest ecosystems. This confines any general conclusions and equally limits our ability to parametrize tropical forest ecosystems in Earth System Models. Nevertheless, the contribution of tropical forest biomes to the global soil GHG budgets maybe disproportionately altered via potential future increases in N and P availability.

**Declaration on Conflict of Interest.** We declare that there is no conflict of interest.

**Author contribution.** JT and OvS conceptualized the study. OvS established the nutrient manipulation experiment. JT conducted the fieldwork, did data analysis and prepared the manuscript. OvS, PF, and SD provided significant input on the experimental set-up and data analysis. RH and BM did laboratory measurements and gave critical feedback on the manuscript. OvS, PF, SD, MG, and LFT critically reviewed and gave feedback on the manuscript.

**Data availability.** Data is available on request.

**Acknowledgement.** We thank F. Babweteera and Budongo Conservation Field Station management for hosting the nutrient manipulation experiment, providing us with working space and the climatic data. Special thanks goes to Johan Six's laboratory ETH-Zürich for analyzing the gas samples. We are grateful to the German Academic Exchange (DAAD) (grant number: 57381412) for JT's stipendium and meeting his travel costs between Uganda and Germany. We also thank the International Foundation of Science (IFS), Stockholm, Sweden, for the financial support (grant number: D/6293-1) towards JT's fieldwork in Uganda, and the National Agricultural Research Organization (NARO) for the institutional support and administration of the IFS grant. We thank the DFG funded Emmy Noether Junior Research Group "TropSOC" (Gepris - project number 387472333) for the additional support towards this study and the DFG funded Individual Research project (RELIANCE; grant number STR 1375/1-1) for setting up the nutrient manipulation experiment. Lastly, we thank G. B. Ayo and M. Adriko for supporting our field measurements.

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

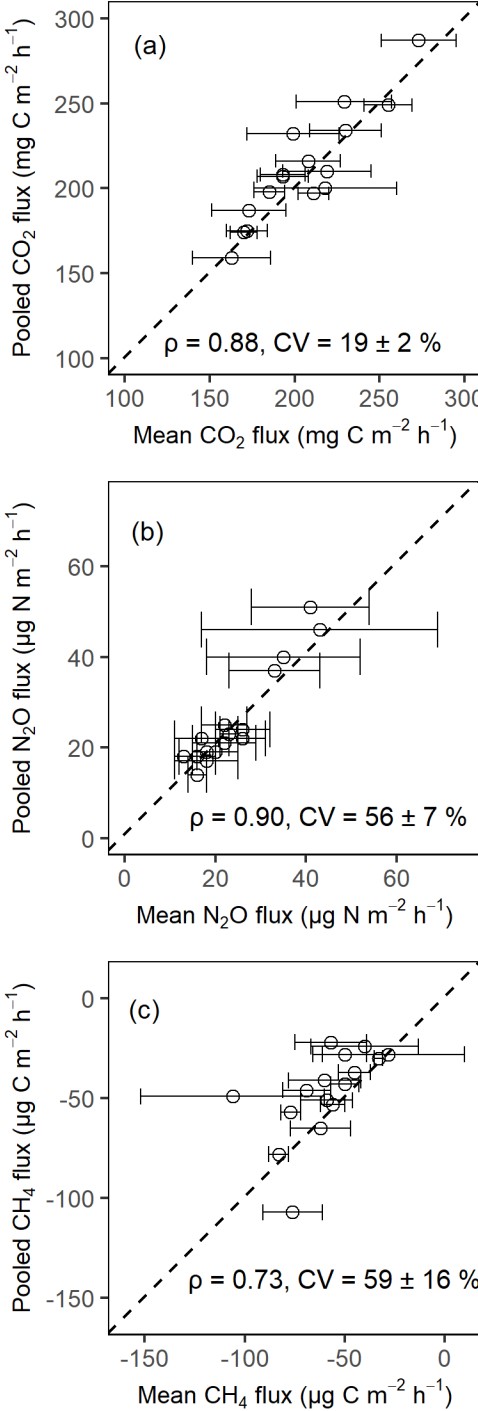

Figure A1. Comparison of the soil $CO_2$ fluxes (a) soil $N_2O$ fluxes (b), and soil $CH_4$ fluxes (c) from pooled sampling and the mean of four chamber measurements for the month of February 2020 in Budongo Forest reserve. $\rho$ is the spearman correlation coefficient, and CV is the coefficient of variation. Error bars are derived from standard error of the mean.