# Peer review of "Nutrient limitations regulate soil greenhouse gas fluxes from tropical forests: evidence from an ecosystem-scale nutrient manipulation experiment in Uganda"

_SOIL, 2020_

## Author Comment (AC1)

**Reviewer_#1 (Comments to Author)**

Dear Reviewer #1, Thank you for the time investment in reading our manuscript and making suggestions for its improvement. We have incorporated the suggestions and made changes accordingly. These are reflected in our point-by-point reply below in blue.

1. It is not clear in the abstract or materials and methods that the fertilizer was split applied. I suggest that the authors work on making this clear from the reader upfront. I also mention the rates and frequency of fertilizer application.

   Author's response:

   We would like to clarify that nitrogen was applied at a rate of 125 kg N ha$^{-1}$ yr$^{-1}$ as urea $((NH_2)_2CO)$ and phosphorous at a rate of 50 kg P ha$^{-1}$ yr$^{-1}$ as triple super phosphate $(Ca(H_2PO_4)_2)$ which is already mentioned in the materials and methods section (see LN 121 to 122 of the original manuscript). Likewise, we also mention that the fertilizers were split into four equal doses annually. This means that, 31.25 kg N ha$^{-1}$ yr$^{-1}$ and 12.5 kg P ha$^{-1}$ yr$^{-1}$ were applied either individually (N or P) or in combination (N + P) to the replicate plots of the nutrient manipulation experiment on a quarterly basis (every three months). We will subsequently include the quarterly rates in the materials and methods sections of the revised manuscript as suggested by the reviewer.

2. Ln 48: The word "geochemsitsry" was supposed to be "geochemistry"

   Author's response:

   Indeed, the word geochemistry was misspelt and has been corrected in the revised manuscript.

3. Ln 121-122: why where these fertilization rates chosen? What was the rationale?

   Author's response:

   The rationale behind these application rates was to produce comparable results with other nutrient manipulation experiments (NME) in the humid tropics. The table below shows the fertilizer application rates used in other ongoing NMEs across the tropics.

   *Table 1. Ongoing nutrient manipulation experiments in tropical forest ecosystems*

   | Site name | Country | N (kg N ha$^{-1}$ yr$^{-1}$) | P (kg P ha$^{-1}$ yr$^{-1}$) |
   |---|---|---|---|
   | **This study** | **Uganda** | **125 - Urea** | **50 - Triple super phosphate** |
   | Gigante | Panama | 125 - Urea | 50 - Triple super phosphate |
   | NITROF | Panama | 125 - Urea | - |
   | EFFEX | Costa Rica | 100 - Urea | 47 - Triple super phosphate |
   | Sabah | Malaysia | 100 - Urea | 50 - Triple super phosphate |
   | Nouragues | French Guiana | 125 - Urea | 50 - Triple super phosphate |
   | Paracou | French Guiana, | 125 - Urea | 50 - Triple super phosphate |
   | AFEX | Brazil | 125 - Urea | 50 - Triple super phosphate |

   Specifically, our experiment was all modeled after the Gigante nutrient manipulation experiment in Panama (Wright et al 2011) and accordingly, we applied the same types of fertilizer, the same fertilizer quantities and applied them in four equal doses.

The premise of these fertilization rates was to create nutrient enriched conditions through application of a relatively large dose of N (125 kg N ha$^{-1}$ yr$^{-1}$), and P (50 kg P ha$^{-1}$ yr$^{-1}$), and measure the ecosystem response including the soil greenhouse gas fluxes. The fertilizer application rates used in our study represent 92 % (136 kg N ha$^{-1}$yr$^{-1}$) of annual N inputs and about 470 % (according to Wright et al., 2011) of the annual P inputs from tropical forests' litter. For ecosystem-scale studies premised in the tropics, it is very important to use a large P dose relative to the annual P litter input in order to overcome the known strong P fixation capacity of tropical soils (Yavitt et al., 2010).

4.  Ln 125: How much fertilizer was applied at different times. Consider mentioning the rates of N and P application.

    Author's response:

    We applied 31.25 kg N ha$^{-1}$ yr$^{-1}$ and 12.5 kg P ha$^{-1}$ yr$^{-1}$ either individually (for N or P) or in combination (N + P) to the replicate plots of the nutrient manipulation experiment each quarter (every three months). Hence, the fertilizers were applied four times each year. As suggested, we will write in the revised manuscript that:

    *"31.25 kg N ha$^{-1}$ yr$^{-1}$ and 12.5 kg P ha$^{-1}$ yr$^{-1}$ were added to the replicate plots of the nutrient manipulation experiment each quarter (every three months) either individually (for N or P) or in combination (N + P)."*

5.  Ln 127-128: How was the soil sampled collected in a pit? Or using augers? Be clearer on what was done.

    Author's response:

    Soil samples were taken at ten different locations within the plot for the 0-10 cm depth. Here, soil monoliths (20 cm (L) x 20 cm (W) x 10 cm (D)) were carefully taken out using a spade. For the depths; 10-30 cm, and 30- 50 cm, soil samples were obtained from five of the ten different locations within each plots using an auger. This will be clearly stated in the methods section of the revised manuscript.

6.  Ln 142: Considering the expected peak in GHG emission following fertilizer application, why was the intensity of GHG monitoring not increased immediately after fertilization?

    Author's response:

    It is a known fact that addition of N will increase N$_2$O fluxes immediately after fertilization (short-term response), but this was not the aim of the study. The aim was to evaluate the long-term effects of N enrichment on ecosystem response (including soil greenhouse gas fluxes). The long-term response reflects the new equilibrium established with elevated N levels. It is also against this background that the respective GHG flux and soil-environmental control datasets were divided into short-term response (transitory phase, < 28 days from fertilization) and long-term response (background phase, > 28 days after fertilization) in order to tease apart the short-term and long-term responses to N addition.

7.  Ln 152: The gas measuring window 9 am-4 pm is too wide. Wouldn't air temperature be different at 9 am and at 3 pm for instance?

Author's response:

While temperature plays an important role in regulating GHG fluxes in the soil, the diurnal air temperature variability at this tropical forest was minimal ($0.6 \pm 0.04$ °C; mean $\pm$ SE). Correspondingly, soil temperatures also had minimal diurnal variability ($0.2 \pm 0.03$ °C; mean $\pm$ SE). In addition, we de-trended any effects of diurnal temperature effects may have on the soil GHG fluxes by randomly selecting the plot to be measured. This ensured that all the plots had an equal chance of being measured either in the morning or mid-afternoon or late afternoon. We are therefore confident that the negligible diurnal variation in both air and soil temperature during the time of sampling did not affect the measured soil GHG fluxes (this can also be seen in Fig. 1).

[Figure]

*Figure 1. Linear relationship between the measured soil greenhouse gas fluxes ($CO_2$, $CH_4$, and $N_2O$) and the time of sampling from the control plots of the NME premised in Budongo Forest Reserve.*

8. Ln 173 Ammonia or Ammonium?

Author's response:

This was a typo. We wrote ammonia in the text but it is supposed to be ammonium. This has been corrected in the revised manuscript.

9.  Ln 264: In Fig 3a it does not appear $CO_2$ fluxes ever went above 250 mg C m$^{-2}$ h$^{-1}$ yet here you give the range as 60 to 330 mg C m$^{-2}$ h$^{-1}$? Please explain or correct.

Author's response:

Referring to Fig 3a was erroneous and an oversight on the part of the authors because the values mentioned in the text were ranges and not means (presented in Fig 3a). This has been corrected.

10. Ln 288: In Fig 3b, it does not appear $CH_4$ uptake was ever above -200 mg C m$^{-2}$ h$^{-1}$, yet here you have it as -278 mg C m$^{-2}$ h$^{-1}$? Please explain or correct.

Author's response:

Referring to Fig 3b was equally erroneous and an oversight on the part of the authors because the mentioned values in the text were ranges and not means (presented in Fig 3b). This has also been corrected in the text.

11. Ln 353: I think "mirobial" was supposed to be "microbial"

Author's response:

This was a typo and has been corrected in the revised version of the manuscript.

12. Ln 400-402: Does the relationship not depend on the form of mineral N (NH4+ of NO3-)? Also, see: Banger, K.; Tian, H.; Lu, C. Do nitrogen fertilizers stimulate or inhibit methane emissions from rice fields? Glob. Chang. Biol. 2012, 18, 3259–3267; for insights on the mechanisms.

Author's response:

Yes, whereas it has been shown that the relationship between $CH_4$ uptake and soil mineral N depends the form of N ($NO_3^-$ or $NH_4^+$), the proportion of $NO_3^-$ and $NH_4^+$ in the soil is key in shaping this relationship. In terrestrial ecosystems, the proportion of the forms of mineral N is influenced by the type of fertilizer used (urea or ammonium based fertilizer) and the nature of the N cycle. The dominant ecosystem N cycles are either closed (soil mineral N dominated by $NH_4^+$ compared $NO_3^-$, Hassler et al., 2015) or open/leaky (soil mineral N dominated by $NO_3^-$ compared $NH_4^+$, e.g. at our study site and that of Koehler et al., 2009). In either N cycles (i.e. closed or open), aggregation of the datasets for the different forms of mineral N during statistical analysis is very common (see Hassler et al., 2015). We used the same approach in our statistical analysis because the $NO_3^-$ ion concentration in the soil at our tropical forest site was significantly and consistently higher than the $NH_4^+$ ions throughout the gas measurement campaign. It is also imperative to note that $CH_4$ uptake was mainly limited by $NO_3^-$ ion content but there was no meaningful correlation between $CH_4$ uptake and $NH_4^+$ content. Similarly, even after aggregation of the $NO_3^-$ and $NH_4^+$ contents together (to get total soil mineral N), the relationship between $CH_4$ uptake and soil mineral N still went in the direction of $NO_3^-$.

13. Ln 430: What do the results look like when you correlate N2O with either NH4+ or NO3-

Author's response:

There was a relatively weak negative correlation between $N_2O$ and $NO_3^-$, and no correlation at all between $N_2O$ and $NH_4^+$. The correlation between $CH_4$ uptake and soil mineral N ($NO_3^-$ content plus $NH_4^+$ content)

was still negative like it were in the case of $NO_3^-$, simply because $NO_3^-$ ions dominated the soil mineral N at our tropical forest study site.

References:

1. Hassler, E., Corre, M. D., Tjoa, A., Damris, M., Utami, S. R., & Veldkamp, E. (2015). Soil fertility controls soil–atmosphere carbon dioxide and methane fluxes in a tropical landscape converted from lowland forest to rubber and oil palm plantations. *Biogeosciences*, *12*(19), 5831-5852.

2. Lukwago, W., Behangana, M., Mwavu, E. N., & Hughes, D. F. (2020). Effects of selective timber harvest on amphibian species diversity in Budongo forest Reserve, Uganda. Forest Ecology and Management, 458, 117809.

3. Koehler, B., Corre, M. D., Veldkamp, E., Wullaert, H., & Wright, S. J. (2009). Immediate and long-term nitrogen oxide emissions from tropical forest soils exposed to elevated nitrogen input. *Global Change Biology*, *15*(8), 2049-2066.

4. Wright, S. J., Yavitt, J. B., Wurzburger, N., Turner, B. L., Tanner, E. V., Sayer, E. J., ... & Corre, M. D. (2011). Potassium, phosphorus, or nitrogen limit root allocation, tree growth, or litter production in a lowland tropical forest. Ecology, 92(8), 1616-1625.

5. Yavitt, J. B., Harms, K. E., Garcia, M. N., Mirabello, M. J., & Wright, S. J. (2011). Soil fertility and fine root dynamics in response to 4 years of nutrient (N, P, K) fertilization in a lowland tropical moist forest, Panama. Austral Ecology, 36(4), 433-445.

---

## Author Comment (AC2)

**Reviewer #2 (Comments to Author)**

1. **General comments:**

1.1. This manuscript presents the results of large-scale nutrient manipulation experiment in a tropical forest in Uganda. Four treatments were considered in this experiment including an unamended control and three different nutrient applications (N, P, and N+P). Greenhouse gas fluxes and other soil data were collected over a fourteen-month experiment. The findings of this manuscript will help advance our understanding of GHG fluxes in African tropical forest ecosystems and how these ecosystems may respond to increases in nitrogen and phosphorus availability.

Author's response:

We thank Reviewer #2 for both the general and specific comments on the manuscript as these have helped us further improve the clarity and overall quality of our manuscript. In addition, we appreciate the reviewer's acknowledgement of the tangible contribution our study makes towards the better understanding of the tropical forest responses to changes in ecosystem nutrient dynamics (particularly nitrogen and phosphorus). Below (in blue) are our point-by-point responses to Reviewer #2 comments.

1.2. (Part 1 of 4) However, the experimental design is vague and needs additional clarifications.

Author's response:

This nutrient manipulation experiment (NME), uses a completely randomized design where N, P and K were applied individually and in all possible combinations (N, P, K, N+P, N+K, P+K, N+P+K), and compared with an unamended control plot. Each of the eight treatments was replicated four times (hence, n = 32 plots; 8 treatments x 4 replications). All plots were established in a compact geographical area where soil properties (physical, chemical and moisture regimes) were similar. The completely randomized design was the most appropriate for this ecosystem-scale NME because the NME involved only a single independent variable — the macronutrients, and several response variables (ecosystem processes). All the treatments were randomly assigned to the experimental units (plots) in order to minimize any possible confounding between the desired treatment effects (macronutrients) and other unknown effects. This experimental design is both statistically sound and also robust to measure the effect of macronutrients (independent variable) on ecosystem processes (dependent/response variables). The study on soil greenhouse gas fluxes — the basis for this manuscript, was conducted on only N, P, N+P, and the unamended control treatment plots (n = 16 plots) because N and P availability has been demonstrated to alter soil greenhouse gas fluxes from tropical forest biomes.

For the benefit of clarity, the text below and in double quotation marks will be added to the experimental design subsection of the materials and methods section of the revised manuscript as follows:

*"The study was carried out in the framework of a running nutrient manipulation experiment (NME). The NME used a completely randomized design to investigate how the three macronutrients (applied individually (N, P, K) and in all possible combination (N+P, N+K, P+K, N+P+K) as treatments) constrained key ecosystem processes (particularly nutrient cycling, net primary productivity, carbon sequestration, and soil greenhouse gas fluxes) in comparison to the unamended control. Each of the eight*

*treatments was replicated four times (hence, n = 32 plots; 8 treatments x 4 replications). However, the soil greenhouse gas study—the basis for this manuscript,  was conducted on only N, P, N+P, and the unamended control treatment plots (n = 16 plots) because N and P availability have been shown to limit soil greenhouse gas fluxes from tropical forest biomes."*

1.2 (Part 2 of 4): The manuscript is also framed as a global change experiment (i.e., increased nutrient deposition), but the amount of N and P applied is not justified and exceeds reasonable nutrient additions in similar ecosystems examining the effects of N and P deposition (e.g., Lu et al. 2018, Van Langehove et al. 2020).

Author's response:

To clarify, this experiment was not designed to simulate the effects of future N deposition on greenhouse gas fluxes. As the reviewer notes, the nutrient application rates used in this study far exceed any realistic future N deposition for this relatively remote area of central Africa. Instead, the aim of our study was to learn how macronutrients regulate background (long-term) soil greenhouse gas fluxes, and specifically to identify the role these macronutrients have in soil GHG production and consumption when ecosystem nutrient limitations are alleviated. These application rates (125 kg N ha$^{-1}$ yr$^{-1}$ and 50 kg P ha$^{-1}$ yr$^{-1}$) are in line with all other NMEs currently ongoing across the tropics (see our response to Reviewer #1 and Table 1) which aim to understand constraints regulating ecosystem processes.

*Table 1. Ongoing nutrient manipulation experiments in the tropical forest ecosystems*

| Site name | Country | N (kg N ha$^{-1}$ yr$^{-1}$) | P (kg P ha$^{-1}$ yr$^{-1}$) |
|---|---|---|---|
| **This study** | **Uganda** | **125 - Urea** | **50 - Triple super phosphate** |
| Gigante | Panama | 125 - Urea | 50 - Triple super phosphate |
| NITROF | Panama | 125 - Urea | - |
| EFFEX | Costa Rica | 100 - Urea | 47 - Triple super phosphate |
| Sabah | Malaysia | 100 - Urea | 50 - Triple super phosphate |
| Nouragues | French Guiana | 125 - Urea | 50 - Triple super phosphate |
| Paracou | French Guiana, | 125 - Urea | 50 - Triple super phosphate |
| AFEX | Brazil | 125 - Urea | 50 - Triple super phosphate |

We have carefully adjusted  the wording in LN 89 and 90 of the original manuscript to ensure that readers are aware this is not an N deposition simulation experiment. LN 89 and 90 will now read as follows:
*"However, a NME study in an African tropical forest would offer valuable insights on the soil GHG flux feedbacks of these understudied biomes in case of alleviations of N and P limitations."*

1.2 (Part 3 of 4) I also have concerns about the greenhouse gas sampling frequency and the time between sample collection and measurement in the lab.

The goal of this study was to evaluate long-term effects of N and P additions, rather than the short-term peaks caused by fertilization or precipitation events/episodes. It is the long-term measurements (made >28 days after fertilization), that represent the new equilibrium established with elevated N or (and) P levels that are particularly relevant for the objectives of this study. A similar approach was reported in the Köhler et al. (2009) study.

Next, monthly measurements are very common when measuring GHG fluxes in the tropics, as they give a relatively high data resolution of temporal trends through the year, while balancing the expenses of the fieldwork. Monthly measurements have been reported in numerous publications in NMEs (e.g. Köhler et al., 2009) and in other GHG studies across the tropics (e.g. Iddris et al., 2020, Hassler et al., 2015, Hassler et al., 2017, Lontsi et al., 2020).

Finally, Labco Exetainers® with the Labco Grey Chlorobutyl Septum can reliably store gas samples for many months before measurement on the gas chromatography. According to Hassler et al. (2015), these Exetainers could reliably store gas samples for periods of up to six months. In our study, all collected gas samples were analyzed within 12 weeks of collection. Furthermore, all plastic caps were screwed on to the gas vials/Exetainers by hand and 'quarter turned' to ensure that they were all airtight (this procedure in outlined in the methodological paper of Pavelka et al. (2018)).

1.2 (Part 4 of 4) And, in general, the primary findings of the experiment are not effectively placed into the context of global changes and the consequences of increasing reactive nitrogen in the environment. Van Langenhove, L., Verryckt, L.T., Bréchet, L. et al. Atmospheric deposition of elements and its relevance for nutrient budgets of tropical forests. Biogeochemistry 149, 175–193 (2020). https://doi.org/10.1007/s10533-020-00673-8 Lu, X., Vitousek, P. M., Mao, Q., Gilliam, F. S., Luo, Y., Zhou, G., ... & Mo, J. (2018). Plant acclimation to long-term high nitrogen deposition in an N-rich tropical forest. Proceedings of the National Academy of Sciences, 115(20), 5187-5192.

> Author's response:
>
> As addressed earlier, the objectives of this study were not to simulate N deposition processes or highlight the cascade of effects this reactive N addition may have on soil greenhouse gas fluxes. Instead, our objective (as stated in the Introduction, LN 91-92 of the original manuscript) was to explore the role elevated background N and P availability has in driving soil GHG fluxes when different nutrient limitations in this ecosystem are lifted. While it may appear minor, we believe there is a very important distinction, which accordingly affects the results we report. Furthermore, the nutrient application rates far exceed any potential future N deposition so that drawing conclusions on N deposition is, in our opinion, not correct.

**Specific comments**

2. General abstract comments:
2.1. Consider framing this experiment in the context of global changes, i.e., increased N and P deposition in natural ecosystems. It is not clear from the initial framing if this study concerns managed forests or native forest ecosystems. It is later explained that the experiment occurred in a forest reserve, and this should be clarified for the reader. LN 30: Listing p-values to three significant figures unnecessary. Consider reducing to two significant figures and changing elsewhere in the text.

Author's response:

As indicated in section 1.2, the aim of the study was not to simulate elevated atmospheric N and P deposition, but instead to investigate the role nutrients have in soil GHG production and consumption when nutrient limitations are alleviated. Atmospheric N and P deposition rates over our tropical forest site have been very low (Galloway et al., 2004), remain low based on our onsite measurements (8.5 kg N ha$^{-1}$ yr$^{-1}$, and 0.03 kg P ha$^{-1}$ yr$^{-1}$), and are expected to only marginally increase in the next 30 years (to about 10 kg N ha$^{-1}$ yr$^{-1}$) (Galloway et al., 2004). It is for this reason that we framed our study as a macronutrient enrichment experiment rather than an N or (and P) deposition simulation study. Furthermore, as Reviewer #2 rightly observed, the fertilizer application rates used in our study would be too high for an N or P deposition simulation study. If our objective had been to simulate atmospheric N and P deposition, we would have applied far lower quantities of nutrient to reflect more realistic future deposition. Fertilizer application rates we used in our study were in line with almost all ongoing NMEs in the tropics currently investigating ecosystem responses to nutrient limitations (see Table 1).

Following the reviewer's advice, we will include upfront (in the abstract) that the study was conducted in a forest reserve, and all the p values used in the text will be reduced to two significant values in the revised version of the manuscript.

3.  General introduction comments:

3.1. The impacts of climate change and alterations to the global N and P cycle should be discussed to contextualize this work, particularly in relation to changing N and P dynamics in forested ecosystems. The authors present other NMEs in tropical forests and the lack of experimentation in tropical Africa, but these studies were largely conducted to understand forest responses to N and P deposition. While the authors mention N deposition in LN 96, this global change driver is not presented earlier in the text, and it is an important consideration and rationale for this work.

Author's response:

We would like to clarify that this was not N or and P deposition simulation study but rather an ecosystem scale study underpinning the soil greenhouse gas response from tropical forest biomes following lifting N or (and) P limitations on soil microbial communities. Consistent with the aim of the study, a strong nutrient pulse (in form of fertilization) was introduced to the forest ecosystem and the soil greenhouse gas fluxes (among other ecosystem responses) measured on a monthly basis. In this respect, we believe that both the literature review and subsequent contextualization of our study were thoroughly done.

3.2. LN 96: What about phosphorus? Please provide additional justification for how changes in P deposition could impact tropical forest and GHG budgets.

Author's response:

We think that we adequately dealt with the effect of P availability on soil GHG budget in the preprint version (please see LN68-LN69 of the pre-print version of the manuscript). However, following the reviewer's advice, we have elaborated on how P availability further opens up the N cycle in the revised manuscript. We will additionally write in the revised manuscript as follows:

*"P availability opens up the N cycle by stimulating increased mineralization of soil organic matter availing more N for soil nitrification or (both) denitrification processes (Mori et al., 2010)".*

3.3. LN 104: Why would P stimulate N release from organic matter? This is mentioned, but not described in detail, in LN 75-84. Perhaps part of my confusion is from the use of organic matter. Do the authors mean soil organic matter or litter? These terms are used interchangeably in LN 81-84.

Author's response:

We meant to say that P availability has been shown to stimulate increased mineralization of soil organic matter availing more N for soil nitrification or (both) denitrification processes (Mori et al., 2010). For clarity, the details in double quotation in comment 3.2. will be added to revised manuscript .

Next, we would also like to indicate that mineralization was used with respect to soil organic matter and not litter. This will be adjusted in the revised manuscript.

4. General methods comments:

4.1. The materials and methods section needs substantial clarifications, including: the rationale for the treatment application rates, when the applications occurred over the course of the experiment, details about the experimental design, and clarification about the GHG flux measurements. Please refer to the detailed comments below.

Author's response:

As suggested by reviewer #2, we will clarify the rationale of the treatment application rates, the timing of the split fertilizer doses, and GHG flux measurements in the text of the methods section of the revised manuscript.

Specifically, we will write in the revised manuscript:

*"The study was carried out in the framework of a running nutrient manipulation experiment (NME). The NME used a completely randomized design to investigate how the three macronutrients (applied individually (N, P, K) and in all possible combination (N+P, N+K, P+K, N+P+K) as treatments) constrained key ecosystem processes (particularly nutrient cycling, net primary productivity, carbon sequestration, (and) soil greenhouse gas fluxes) in comparison to the unamended control. Each of the eight treatments was replicated four times (hence, n = 32 plots; 8 treatments x 4 replications). However, the soil greenhouse gas study — the basis for this manuscript, was conducted on only N, P, N+P, and the unamended control treatment plots (n = 16 plots) because N and P availability have been shown to limit soil greenhouse gases from tropical forest biomes. Each treatment plot measured 40 m x 40 m in size with an inner core measurement zone (30 m x 30 m) to avoid boundary effects. A spacing of at least 40 m between experimental plots was ensured to prevent spillover of applied nutrients from the neighboring plots. To achieve N and P enriched conditions, Nitrogen was applied at a rate of 125 kg N $ha^{-1}$ $yr^{-1}$in form of urea ($(NH_2)_2CO$), and P at 50 kg P $ha^{-1}$ $yr^{-1}$as triple super phosphate ($Ca(H_2PO_4)_2$), with these fertilizers split into four dozes annually. The fertilizer application rates used in this study represent 92 % (136 kg N $ha^{-1}yr^{-1}$) of annual N inputs and about 470 % (according to Wright et al., 2011) of the annual P inputs from the litter. Additionally, the rates were comparable to those used in Wright et al. (2011) allowing us to stretch our conclusions beyond the Ugandan tropical forest site".*

4.2. LN 113: Please use a more appropriate citation. The authors might consider the WorldClim dataset.

Author response:

The citation has been changed to Lukwago et al. (2020) because their study was conducted in Budongo Forest reserve and they reported an average temperature (of about 25 °C) and precipitation (of about 1700 mm) for this region.

4.3. LN 121-125: Additional information about the NME needs to be described. Please add a citation if one exists of previously published work from this site. At a minimum, the text should provide additional clarification regarding the experimental design, i.e., was it randomized? It is also unclear what the number of replicates is in each treatment. Please include in the text that there were four blocks or four replicated plots per treatment.

Author's response:

Additional information about the NME will be provided (as highlighted in the response to comment 4.1.) including the detailed description of the experiment (i.e. completely randomized experimental design, consisting of eight treatments, with each treatment replicated four times). None of the work from the study site has been published yet.

4.4. LN 127-128: The nitrogen and phosphorus additions rates need justification. These rates are unusually high for N and P deposition experiments, and the rates align more closely with those common in agricultural fertilization experiments. This is one of my primary concerns with the framing of this experiment; the applications rates seem far too high to justify as N or P deposition.

Author's response:

As stated in comment 2.1 and 3.1, we would like to reiterate that the aim of the study was to understand how soil greenhouse gas fluxes respond to macronutrient enrichment in tropical forests and not simulate gradual effects of N or P deposition on soil GHG fluxes. Accordingly, we created an N and P enriched environment in this tropical forest by applying a relatively large dose of N (125 kg N $ha^{-1}$ $yr^{-1}$ as urea) and P (50 kg P $ha^{-1}$ $yr^{-1}$ as triple super phosphate). The N and P fertilization rates of 125 kg N $ha^{-1}$ $yr^{-1}$ and 50 kg P $ha^{-1}$ $yr^{-1}$ represented 92 % (136 kg N $ha^{-1}$ $yr^{-1}$) about 470 % (according to Wright et al. (2011)) of the annual N and P inputs in the litter respectively at our tropical forest site. It is worth mentioning that for ecosystem-scale studies premised in the tropics, a large P dose is applied relative to the annual P litter input in order to overcome the known strong P fixation capacity of tropical soils (Yavitt et al., 2011). The fertilizer application rates used in our study have not only been used by Wright et al. (2011) but also in ongoing NMEs listed in Table 1.

4.5. LN 135: How were these soil samples collected, i.e., shovel or core?

Author's response:

In every plot, soil samples from 0 - 10 cm depth were collected using a spade, which we used to remove soil monoliths (20 x 20 x 10 cm) from 10 randomly located spots per plot. These soil monolith samples were subsequently mixed together in a large basin, from where we removed approximately a 500 g homogenized soil sample for laboratory analysis. For deeper soils (10 - 30, and 30 - 50 cm), soil samples

were obtained from five of the 10 sampling locations using a heavy-duty gouge auger. Here too we collected a composite (pooled) sample for each respective depth.

4.6. LN 148: I have concerns regarding this sampling frequency and the subsequent calculations of GHG annual fluxes. This measurement frequency is far too coarse to capture the sensitivity of N2O to precipitation events. From Figure 2, it appears like there were many pulses in precipitation over the experimental period, which may have resulted in substantial N2O release. While I acknowledge the difficulty in sampling at a twice weekly or weekly sampling frequency, the manuscript should describe why this monthly interval was selected for measurement.

Author's response:

While we recognize that the reviewer is correct, and that by only measuring monthly we may not capture the small-scale variability in $N_2O$ fluxes (including some precipitation induced flushes), our interest in this study was to observe longer-term background controls, namely how nutrient availability regulates GHG fluxes. It is for the same reason that we divided the soil GHG flux dataset into transitory and background phases during statistical analysis in order to tease apart the immediate responses to fertilization from the long-term ones. The transitory phase included all measurements taken between 0-28 days following fertilization while background fluxes included all the GHG fluxes measured more than 28 days from fertilization (i.e. after the disappearance of the fertilization induced GHG flushes/peaks). As mentioned above in response to comment 1.2 (part 3 of 4), the same sampling frequency has been used in many other studies in the humid tropics (Iddris et al., 2020, Hassler et al., 2015, Hassler et al., 2017, and Lontsi et al., 2020).

4.7. LN 151-152: Was litter/residue left inside the chamber or was the soil kept bare?

Author's response:

Litter/residue was left inside the chamber. However, the chamber was always maintained vegetation free in order to avoid measuring night respiration of the plants during chamber closure.

4.8. LN 149-150: I have concerns about the area of the chambers and the sampling times used in this experiment. Carbon dioxide fluxes are usually orders of magnitude greater than N2O or CH4; a larger chamber area is usually necessary to estimate these fluxes from soil. Furthermore, while the sampling times for N2O and CH4 make sense, I am concerned that CO2 may have plateaued during this interval, impacting CO2 diffusion, and the CO2 concentration measured. Did the authors test for a linear relationship in their pooled and unpooled approach? How representative do the authors feel the chambers were of the overall plot GHG fluxes given the small size of these chambers?

Author's response:

Chamber design: Up until now, there is still no standard chamber design because the design of the chamber is dictated by the nature of the ecosystem under investigation (Pavelka et al., 2018). In this methodological paper, Pavelka et al. (2018) indicated that the chamber design should at least cover a minimum ground area of 0.2 m x 0.2 m (0.04 m$^2$). In our experiment, we used a circular chamber with a diameter of 0.237 m that covered a ground area of 0.044 m$^2$. Therefore, our chamber design was well

within the recommendation confines of Pavelka et al. (2018). Koehler et al. (2009) and Matson et al. (2017), too, used a similar chamber design to measure soil greenhouse gas fluxes (including carbon dioxide fluxes) from their tropical forest sites.

Sampling times used in this experiment: Our sampling times were consistent with those used in the separate studies of Koehler et al. (2009) and Matson et al. (2017) to estimate soil carbon dioxide (among other soil greenhouse gas) fluxes from tropical forest ecosystems. Moreover, in all these studies, the used sampling times were well below the 45-minute maximum chamber closure period recommendation by Pavelka et al. (2018). Wanyama et al. (2019) used the 45-minute maximum chamber closure period and still obtained decent estimates for soil $CO_2$ fluxes. They did not report any incidences of plateauing in the measured $CO_2$ fluxes. Furthermore, we inspected the linear increase of $CO_2$ concentration during chamber closure for all the batches of gas samples as a quality control check. Here, the $R^2$ was typically above 0.95, with no evidence of plateauing.

Linear relationship in their pooled and unpooled approach: We collected both pooled and unpooled gas samples for the month of February 2020 and tested how the pooled approach compared to the unpooled for all the three gases. There was no significant difference in the concentration of all the three gases between the pooled and unpooled approaches (see Fig. 1 below). We also mentioned this in the original manuscript (see LN 150) but we will include this figure as supplementary material for the revised manuscript.

[Figure]

Figure 1. Comparison of the soil $CO_2$ fluxes (a), soil $CH_4$ fluxes (b), and soil $N_2O$ fluxes (c) measured with the pooled and unpooled approaches during the month of February 2020 in Budongo Forest reserve.

How representative were the measurements given the size of the chamber used: We are confident that the flux measurements presented in our manuscript are representative of the fluxes from the different plots of the NME in this tropical forest biome. This is because our chamber design was in conformity with the minimum ground area requirement for representative terrestrial GHG flux measurements (Pavelka et al., 2018). Additionally, we minimized any plot level spatial variability by randomly deploying the four chambers within the plot. All the pooled gas samples in every plot (at each time interval during chamber closure) were always a composite of the respective head air spaces of the four chambers.

4.9. LN 159: The duration between sample collection and measurement needs additional information. How long was the duration between sample collection and measurement? While generally stable for period of days to a couple of weeks, exetainers are not ideal for long-term storage of gas samples, which should ideally be measured immediately (up to 72 hours) after collection. Please describe the care that was taken to ensure there was no degradation to the gas samples over time.

Author's response:

Potential gas leakages in exetainers are mainly due to the type of exetainers used and the fact that the exetainers are dispatched from the factory with loose caps (not airtight). We purposely selected Labco exetainers (Labco Limited, Lempeter, UK) with screw-on plastic caps fitted with Labco Grey Chlorobutyl Septum for this study because they have been demonstrated to remain airtight for periods spanning up to six months (Hassler et al., 2015). Moreover, all plastic caps were screwed on to the gas vials by hand, and then 'quarter turned' to ensure that they were all airtight (see this procedure in Pavelka et al. (2018)). Although, the Labco exetainers can remain airtight for periods up to six months (Hassler et al., 2015), we ensured that on average, all collected gas samples were analyzed under 12 weeks. We submitted all the samples to the laboratory at ETH Zürich in five batches across the 14 months of field gas sampling.

LN 187: Please provide a citation for this method.

Author's response:

As suggested, we have provided a citation for the method.

4.10. LN 190-204: The manuscript should include additional details about a) the frequency of measurements, chamber size, etc. for the trenching experiment, and b) how the authors portioned CO2 to autotrophic and heterotrophic sources and a citation for their methodology.

Author's response:

As suggested by reviewer #2, additional details about the trenching experiment will be provided in the revised manuscript including the citation for methodology used. We will specifically provide the following additional details in the revised manuscript:

- *"The measurements to disentangle the sources of soil $CO_2$ effluxes spanned over a period of four months (starting in November 2019 and ending in February 2020) and were done on a monthly basis. We purposely selected this measurement period because it represented the transition from the wet season to the long dry season allowing us to capture how soil moisture constrained the different soil $CO_2$ efflux sources.*

- *Both the trenched and reference chamber bases had a design (area = 0.044 $m^2$, and volume = about 12 L) identical to the one used in the nutrient manipulation soil GHG flux study.*

- *The different soil $CO_2$ source were calculated as follows:*

    *Heterotrophic (microbial) respiration = $CO_2$ effluxes from the trenched chamber*

    *Autotrophic (root) respiration = $CO_2$ effluxes from the reference chamber – Heterotrophic respiration".*

4.11. More information about the estimation of root biomass (number of cores, how samples were processed) should also be included, especially because these data are discussed in the results and discussion.

Author's response:

As suggested by the reviewer, the details on estimation of root biomass will be included in the revised version of the manuscript. We will specifically write:

*"Root biomass distribution with depth was determined by digging three profile pits (measuring 1 m x 1 m x 1 m) at the forest site. In every dug pit, ten soil monoliths (each measuring: 20 cm (L) x 20 cm (W)) were carefully cut out (using a spade and hoe) following a 10 cm depth interval from the surface down to 1 m. The soil monolith were thoroughly washed to isolate the roots from the bulk soil. The root samples were oven dried at 60 °C for 48 hours and weighed to determine the root biomass per depth increment. The root biomass for each depth interval was calculated as the mean of the root biomass from the three pits for that interval".*

4.12. LN 212: Is it common to refer to MANOVA as LMEMS?

Author's response:

Not exactly, because multivariate analysis of variance (MANOVA) is used when you have two or more response variables in an experiment and you jointly treat them as one multivariate response variable. The MANOVA structure cannot support multi stratum analysis of variance and its formula is always without an error term. Contrary to the MANOVAs, linear mixed effects models (LMEMs), like majority of the statistical approaches, deal with a single response variable, support multi stratum analysis of variance, and their formula always includes an error term.

4.13. LN 219: A description of the interpolation method used to calculate annual GHG fluxes should be described here. I am also confused why the authors present these data but did not do any statistical analyses with them? If these data are included in the results, they should be analyzed statistically.

Author's response:

As suggested, the description of the interpolation method will be included in the methods section of the revised manuscript. As you will also read in similar studies, for instance; Koehler et al. (2009), Veldkamp et al. (2013), and Iddris et al. (2020), it is recommended that statistical analyses be conducted on only actual (monthly) measurements and not the annual fluxes because the latter are obtained through interpolation of measured actual soil GHG fluxes over the sampling time period. Annual GHG fluxes are however, an important result to present, so as to allow inter-comparability between different forests and experiments.

We will additionally write in the materials and methods section of the revised manuscript that:

*"Annual soil GHG fluxes were obtained through conducting a trapezoidal interpolation on the measured monthly soil GHG fluxes, assuming constant flux rates per day. It is worth mentioning that the annual soil GHG fluxes from the different treatments were not statistically analyzed because they were not actual measurements but rather interpolations."*

4.14. LN 231: Please include the R packages used in the analyses.

Author's response:

As suggested, the key R packages used in the statistical analyses will be added to the methods section of the revised manuscript. As indicated in the original manuscript, statistical analyses were mainly accomplished with linear mixed effects model (LMEMs) and one-way analysis of variance (ANOVA). We used the *'nlme'* and *'car'* packages for LMEMs and one-way ANOVA tests respectively.

We will specifically add a sentence in the revised manuscript stating that:

*"For statistical analyses, we used the 'nlme' and 'car' packages for LMEMs and one-way ANOVA tests respectively".*

5. General results comments: There are several occurrences in tables and figures where analyses are referenced, but they were not described in the text. This information is more appropriate to include at length in the methods section, and it is inappropriate to only provide as footnotes.

Author's response:

As suggested by the reviewer, we will make adjustment in the text of the revised manuscript. See the proposed revised texts (in italics and double quotation marks in comment 4.10, 4.11, and 4.13) to be added to the methods and materials section of the revised manuscript.

5.1. Table 1: If the authors present isotope data, they should describe how these data were collected.

Author's response:

The isotope data presented in this manuscript is from a sister study and we will clarify this by adding a footnote to Table 1.

5.2. Figure 2: Why were these climatic data not used to estimate 30-yr mean annual temperature and precipitation? The use of this weather stations should be described in the methods section.

Author's response:

The weather station data presented in our manuscript was only available for the period of the experiment (about 2 years). This data was beneficial to understand how for instance precipitation constrained soil greenhouse gas fluxes given its direct control on water filled pore space. For the long-term average of the study region, we will cite the Lukwago et al. (2020) study (because they reported long-term temperature and precipitation for this forest site), instead of extrapolating of our 2-year climatic snap shot data to the 30-year climatic average which could be misleading. As suggested by the reviewer, we write in the materials and methods section that:

*"The weather data from our field station was only available for the period of the experiment, and therefore, was used to understand how precipitation constrained soil greenhouse gas fluxes given its direct control water filled pore space".*

6. General discussion comments: I do not find the claim that the ecosystem is "complex" a compelling argument for interpreting the results of the study. The manuscript should omit this language. I also recommend the manuscript include an addition section in the discussion placing the findings of this study in context – how do these results fit into findings of other tropical forest NME and changing N and P

deposition rates in forested ecosystems? The broader impact and relevance to the science and policy communities would strengthen the framing of the manuscript.

Author's response:

We will rephrase the argument on the complexity of the ecosystem in the revised manuscript.

Next, it is evident throughout the discussion section that on top of explaining how the soil GHG production and consumption processes at this tropical forest site were directly or indirectly affected by the alleviation of nutrient limitations, we fit our NME findings in the context of other tropical forest NME (for instance see LN 352-356, LN391-396, LN420-421, LN424-426, e.t.c. of the preprint version of manuscript). Like we already elaborated in the preceding comment sections, we think that introducing a subsection in the discussion, putting our study findings in the context of increasing N or (and) P deposition would be quite misleading. Why? Because the N or P deposition rates over our tropical forest site are quite low. Additionally, the large fertilizer application rates used in our study only served the purpose of achieving a nutrient enriched environment at our tropical forest site to measure a soil GHG flux response. These fertilizer application rates were too high to reflect any realistic projection in N or P deposition over this tropical region.

6.1. LN 357: See previous comments about CO2 measurement and sampling frequency concerns.

Author's response:

Please refer to our response to comment 4.8, for details on the chamber design and the sampling times used in our study.

6.2. LN 433: Please provide additional information about P availability would open the N cycle.

Author's response:

As suggested by the reviewer, additional information on how P availability would open up the N cycle will be added to the revised manuscript. Please see our response to comment 3.3.

7. General conclusion comments: Please clarify the rationale of this experiment: increased nutrient deposition or fertilization for enhanced forest production? Again, all ecosystems are complex, and this is a weak interpretation of the findings of this study.

Author's response:

We sought to understand the soil greenhouse gas flux response of tropical forest biomes under enriched N and P soil conditions and not necessarily simulate the effects of N and P deposition on soil greenhouse gas fluxes from these biomes. Again, the statement on the complexity of the ecosystem has been rephrased in the conclusion section.

References:

1. Hassler, E., Corre, M. D., Tjoa, A., Damris, M., Utami, S. R. and Veldkamp, E.: Soil fertility controls soil-atmosphere carbon dioxide and methane fluxes in a tropical landscape converted from lowland forest to rubber and oil palm plantations, Biogeosciences, 12(19), 5831–5852, doi:10.5194/bg-12-5831-2015, 2015.

2. Hassler, E., Corre, M. D., Kurniawan, S. and Veldkamp, E.: Soil nitrogen oxide fluxes from lowland

forests converted to smallholder rubber and oil palm plantations in Sumatra, Indonesia, Biogeosciences, 14(11), 2781–2798, doi:10.5194/bg-14-2781-2017, 2017.

3. Iddris, N. A.-A., Corre, M., Yemefack, M., van Straaten, O. and Veldkamp, E.: Stem and soil nitrous oxide fluxes from rainforest and cacao agroforest on highly weathered soils in the Congo Basin, Biogeosciences Discuss., 1–46, doi:10.5194/bg-2020-164, 2020.

4. Koehler, B., Corre, M. D., Veldkamp, E. and Sueta, J. P.: Chronic nitrogen addition causes a reduction in soil carbon dioxide efflux during the high stem-growth period in a tropical montane forest but no response from a tropical lowland forest on a decadal time scale, Biogeosciences, 6(12), 2973–2983, doi:10.5194/bg-6-2973-2009, 2009.

5. Lu, X., Vitousek, P. M., Mao, Q., Gilliam, F. S., Luo, Y., Zhou, G., Zou, X., Bai, E., Scanlon, T. M., Hou, E. and Mo, J.: Plant acclimation to long-term high nitrogen deposition in an N-rich tropical forest, Proc. Natl. Acad. Sci. U. S. A., 115(20), 5187–5192, doi:10.1073/pnas.1720777115, 2018.

6. Lukwago, W., Behangana, M., Mwavu, E. N., & Hughes, D. F. (2020). Effects of selective timber harvest on amphibian species diversity in Budongo forest Reserve, Uganda. Forest Ecology and Management, 458, 117809.

7. Matson, A. L., Corre, M. D., Langs, K. and Veldkamp, E.: Soil trace gas fluxes along orthogonal precipitation and soil fertility gradients in tropical lowland forests of Panama, Biogeosciences, 14(14), 3509–3524, doi:10.5194/bg-14-3509-2017, 2017.

8. Mori, T., Ohta, S., Konda, R., Ishizuka, S. and Wicaksono, A.: Phosphorus limitation on $CO_2$, $N_2O$, and NO emissions from a tropical humid forest soil of South Sumatra, Indonesia, ICEEA 2010 - 2010 Int. Conf. Environ. Eng. Appl. Proc., 18–21, doi:10.1109/ICEEA.2010.5596085, 2010.

9. Pavelka, M., Acosta, M., Kiese, R., Altimir, N., Brümmer, C., Crill, P., Darenova, E., Fuß, R., Gielen, B., Graf, A., Klemedtsson, L., Lohila, A., Longdoz, B., Lindroth, A., Nilsson, M., Jiménez, S. M., Merbold, L., Montagnani, L., Peichl, M., Pihlatie, M., Pumpanen, J., Ortiz, P. S., Silvennoinen, H., Skiba, U., Vestin, P., Weslien, P., Janous, D. and Kutsch, W.: Standardisation of chamber technique for $CO_2$, $N_2O$ and $CH_4$ fluxes measurements from terrestrial ecosystems, Int. Agrophysics, 32(4), 569–587, doi:10.1515/intag-2017-0045, 2018.

10. Tchiofo Lontsi, R., Corre, M. D., Iddris, N. A. and Veldkamp, E.: Soil greenhouse gas fluxes following conventional selective and reduced-impact logging in a Congo Basin rainforest, Biogeochemistry, 151(2–3), 153–170, doi:10.1007/s10533-020-00718-y, 2020.

11. Veldkamp, E., Koehler, B. and Corre, M. D.: Indications of nitrogen-limited methane uptake in tropical forest soils, Biogeosciences, 10(8), 5367–5379, doi:10.5194/bg-10-5367-2013, 2013.

12. Wanyama, I., Pelster, D. E., Butterbach-Bahl, K., Verchot, L. V., Martius, C. and Rufino, M. C.: Soil carbon dioxide and methane fluxes from forests and other land use types in an African tropical montane region, Biogeochemistry, 143(2), 171–190, doi:10.1007/s10533-019-00555-8, 2019.

13. Wright, S. J., Yavitt, J. B., Wurzburger, N., Turner, B. I., Tanner, E. V. J., Sayer, E. J., Santiago, L. S., Kaspari, M., Hedin, L. O., Harms, K. E., Garcia, M. N. and Corre, M. D.: Potassium, phosphorus, or nitrogen limit root allocation, tree growth, or litter production in a lowland tropical forest, Ecology, 92(8), 1616–1625, doi:10.1890/10-1558.1, 2011.

14. Yavitt, J., Harms, K., Garcia, M., Mirabello, A. and Wright, S.: Soil fertility and fine root dynamics in response to 4 years of nutrient (N, P, K) fertilization in a lowland tropical moist forest, Panama, Austral

Ecol, 36, 2011.

---

## Author Response (AR1)

**Author response to the Topical Editor**

Comments to the Author:

Both reviewers find the scope of the study to be interesting and worthy to be published in Soil journal. However, they also both express concerns with the need clarity for the methodology, particularly the experimental design and management. They also both raised concerns with the need to justify the high fertilizer. While the authors argue that the rates are in accordance with other similar studies, I don't think that's adequate. I suggest the authors revise the manuscript according to the reviewers comments.

Dear Dr. Pauline Chivenge,

On behalf of all the authors, allow me extend our sincere gratitude to you and both reviewers for finding time to review our manuscript (soil-2020-94) and for their constructive suggestions towards its improvement.

Let me take this opportunity to clarify that the primary objective of this study was to understand the roles nitrogen and phosphorus have in regulating soil GHG fluxes in a humid tropical forest in northwestern Uganda. This experiment was one important aspect of a larger experiment where the overarching aim was to disentangle how nutrients regulate ecosystem processes at a hierarchy of scales from microbial communities to ecosystem carbon accumulation (net primary production). To elicit an ecosystem scale response, and to align our results with previous and ongoing nutrient manipulation experiments (NME) across the tropics, we applied relatively high nutrient doses. Considering these fertilizer doses are quite high, we think that our study does not qualify as a "realistic" N and P deposition simulation study. Nevertheless, we think that our findings can be extended to shed valuable insights on soil GHG flux responses to both abrupt changes in soil nutrient availability (at the onset of the rainy season) and increases in N and P deposition projected for this region. Accordingly, and in line with both your comments and those of the reviewers, we have made the following adjustments in the revised manuscript:

1. Carefully adapted the text in the abstract, introduction and the discussion to integrate the global change aspect into our study. Furthermore, we added a subsection (4.4) in the discussion explaining the implication of our study findings with respect to predicted increases in N and P deposition rates on tropical forest ecosystems.
2. Made substantial clarifications on the experimental design, and explained the rationale behind the fertilization rates used in the study.

We have now addressed the concerns of both reviewers and accordingly incorporated the changes in the revised manuscript. Furthermore, we updated our earlier responses to the reviewer comments, which you can find below. In our response, all the line (LN) references are with respect to the revised manuscript with track changes turned on and not the original version. We hope that the changes we have made in the revised manuscript and our responses to the reviewers satisfy you and the reviewers' concerns. Kindly, do not hesitate to contact me in case you need further clarifications or information about the manuscript.

Yours sincerely,

Joseph Tamale

**Author response to Reviewer_#1**

1. It is not clear in the abstract or materials and methods that the fertilizer was split applied. I suggest that the authors work on making this clear from the reader upfront. I also mention the rates and frequency of fertilizer application.

   Author's response:

   As suggested by the reviewer, we have stated in the materials and methods section of the revised manuscript, that the fertilizer was split in four doses annually, and that 31.3 kg N ha$^{-1}$ and 12.5 kg P ha$^{-1}$ were applied each quarter (every three months). We, however, think that this would be too much information to include in the abstract. Specifically, we added the following text in the revised manuscript (LN 165 to LN 167):

   *"The fertilizer was applied by hand, and in four split dozes every year. Specifically, 31.3 kg N ha$^{-1}$ and 12.5 kg P ha$^{-1}$ were applied to the plots of the NME every three months between May 2018 and June 2020."*

2. Ln 48: The word "geochemsitsry" was supposed to be "geochemistry"

   Author's response:

   Indeed, the word geochemistry was misspelt and has been corrected in the revised manuscript.
   Please see LN 55 for the correction.

3. Ln 121-122: why where these fertilization rates chosen? What was the rationale?

   Author's response:

   This GHG experiment was one important aspect of a larger experiment where the overarching aim was to disentangle how nutrients regulate ecosystem processes at a hierarchy of scales from microbial communities to ecosystem carbon accumulation (net primary production). To elicit an ecosystem scale response, and to align our results with previous (e.g. Wright et al. (2010), Koehler et al. (2009)) and ongoing nutrient manipulation experiments (NME) across the tropics (Table 1), we applied relatively high nutrient doses (i.e. 125 kg N ha$^{-1}$ yr$^{-1}$ as urea and 50 kg P ha$^{-1}$ yr$^{-1}$ as triple superphosphate ).

   *Table 1. Ongoing nutrient manipulation experiments in tropical forest ecosystems*

   | Site name | Country | N (kg N ha$^{-1}$ yr$^{-1}$) | P (kg P ha$^{-1}$ yr$^{-1}$) |
   | --- | --- | --- | --- |
   | **This study** | **Uganda** | **125 - Urea** | **50 - Triple super phosphate** |
   | Gigante | Panama | 125 - Urea | 50 - Triple super phosphate |
   | NITROF | Panama | 125 - Urea | - |
   | EFFEX | Costa Rica | 100 - Urea | 47 - Triple super phosphate |
   | Sabah | Malaysia | 100 - Urea | 50 - Triple super phosphate |
   | Nouragues | French Guiana | 125 - Urea | 50 - Triple super phosphate |
   | Paracou | French Guiana, | 125 - Urea | 50 - Triple super phosphate |
   | AFEX | Brazil | 125 - Urea | 50 - Triple super phosphate |

4. Ln 125: How much fertilizer was applied at different times. Consider mentioning the rates of N and P application.

Author's response:

We applied 31.3 kg N ha$^{-1}$ and 12.5 kg P ha$^{-1}$ either individually (for N or P) or in combination (N + P) to the replicate plots of the nutrient manipulation experiment each quarter (every three months). As suggested by the reviewer, this information has been added to the materials and method section of the revised manuscript. Please see LN 165 to LN 167 of the revised manuscript for the changes.

In the manuscript we added the following text:

*"The fertilizer was applied by hand, and in four split dozes every year. Specifically, 31.3 kg N ha$^{-1}$ and 12.5 kg P ha$^{-1}$ were applied to the plots of the NME every three months between May 2018 and June 2020."*

5. Ln 127-128: How was the soil sampled collected in a pit? Or using augers? Be clearer on what was done.

Author's response:

For the 0-10 cm depth, soil monoliths (20 cm (L) x 20 cm (W) x 10 cm (D)) were carefully taken from ten different locations within each plot of the NME (n = 32 plots) using a spade. However, for deeper soil layers (0-30 cm and 30- 50 cm), samples were obtained outside the established NME plots in order to minimize modifications to the microenvironment inside the NME plots. Deeper soil sampling was done during a reconnaissance survey conducted at approximately 500 m from the current location of the NME site. During the reconnaissance survey, sixteen plots (n = 16) were established and samples taken from five different locations for every depth interval (i.e. 0-30 cm and 30- 50 cm) using an auger (diameter = 30 mm). As suggested by the reviewer, this information has been included in the revised manuscript, please see LN 173 to LN 185 of the revised manuscript. We have specifically written:

*"For the 0-10 cm depth, soil monoliths (20 cm (L) x 20 cm (W) x 10 cm (D)) were carefully taken from ten different locations within each plot of the NME (n = 32 plots) using a spade. However, for deeper soil layers (0-30 cm and 30- 50 cm), samples were obtained outside the established NME plots in order to minimize modifications to the microenvironment inside the NME plots. Deeper soil sampling was done during a reconnaissance survey conducted at approximately 500 m from the current location of the NME site. During the reconnaissance survey, sixteen plots (n = 16) were established and samples taken from five different locations in each plot for every depth interval (i.e. 0-30 cm and 30- 50 cm) using an auger (diameter = 30 mm). The samples from each depth were mixed thoroughly in a basin, and about 500 g of the homogenized samples sent to the soils laboratory of the University of Göttingen, Germany, for analysis."*

6. Ln 142: Considering the expected peak in GHG emission following fertilizer application, why was the intensity of GHG monitoring not increased immediately after fertilization?

Author's response:

We are very thankful for this comment as it allows us to rethink our approach to estimate the short-term effect of N fertilization on N$_2$O fluxes. Only monthly measurements of N$_2$O fluxes were performed for two reasons: (i) overall, we were mostly interested in long-term effects of fertilization on soil GHG fluxes and, (ii) it was challenging to handle large numbers of gas samples at our very remote site in Uganda, so we decided to have a sampling frequency that would enable us answer our main research question, while keeping the number of collected gas samples within our limited handling capacity. However, based on

the comment of the reviewers, we analyzed the effect that time since N fertilization has on $N_2O$ fluxes during the transitory phase (< 28 days from fertilization) to make sure that our results were not biased due to the different timing of GHG sampling after fertilization (Fig. 1). Based on this analysis, we performed a detrending of the measured $N_2O$ fluxes for the transitory phase using the log normal (ln) fit shown in Fig. 1 and eq.1. The adopted $N_2O$ fluxes were then used in the calculation of the $N_2O$ budgets.

$$y = 261.4 - 63.93 \ln(x) - - - - - - - - - - - - - - - - - - - - - - - - - - - - - (1)$$

[Figure]

Figure 1. Log normal relationship between mean $N_2O$ fluxes measured from N addition plots (N, N + P)) and the days since fertilization in the transitory phase.

Table 2. Actual $N_2O$ fluxes measurements and detrended $N_2O$ fluxes measurements (obtained with the log normal fit (eq.1)) from N addition plots (N, and N +P) during the transitory phase

| Days since fertilization | Treatment | $N_2O$ fluxes ($\mu g\ N\ m^{-2}\ h^{-1}$) | |
|---|---|---|---|
| | | Actual measurements | Detrended measurements |
| 5 | N | 145.9 | 93.8 |
| 5 | N+P | 228.9 | 176.8 |
| 12 | N | 18.4 | 31.3 |
| 12 | N+P | 26.1 | 35.3 |
| 13 | N | 115.2 | 124.1 |
| 13 | N+P | 81.6 | 90.6 |
| 14 | N | 82.3 | 96 |
| 14 | N+P | 144.3 | 167 |
| 25 | N | 134.8 | 185.5 |
| 25 | N+P | 35 | 85.8 |
| Mean (± SE; n = 5) $N_2O$ fluxes from N addition plots | | 99.3 ± 23.0 | 106.1 ± 25.0 |
| Mean (± SE; n = 5) $N_2O$ fluxes from N + P addition plots | | 103.2 ± 37.8 | 111.1 ± 26.7 |

The results of the detrending versus the actual $N_2O$ flux measurements in the transitory phase are presented in Table 2. It is interesting to note that, even if we fully agree that detrending was very important to get reasonable $N_2O$ flux estimates for the transitory phase and we very much acknowledge the reviewers' comments here, the overall conclusions from our study were not affected by this adjustment.

In the manuscript, although we briefly added the information below (LN 284 to LN 287) in the methods section, we intentionally did not give more details as we had the impression that this would overrate the importance of the $N_2O$ transitory phase in the analysis compared to the overall analysis of the other GHG measurements in our study. However, if the editor or reviewers think we should elaborate on this more in detail; we would be happy to do so.

*"Prior to statistical analysis, transitory $N_2O$ fluxes from N addition plots (N, and N + P) were detrended to compensate for absence of frequent measurements immediately after fertilization coming from sampling GHGs monthly. Detrending involved using a log-normal fit between the measured $N_2O$ fluxes and time since fertilization (until day 42), and this explained 43 % of the variability in the $N_2O$ data during the transitory phase (p < 0.05)."*

7. Ln 152: The gas measuring window 9 am - 4 pm is too wide. Wouldn't air temperature be different at 9 am and at 3 pm for instance?

   Author's response:

   While temperature plays an important role in regulating GHG fluxes in the soil, the diurnal air temperature variability at this tropical forest was minimal (average ± SE air temperature range during the measurement period: $0.6 \pm 0.04$ °C). Correspondingly, soil temperatures also had minimal diurnal variability (average ± SE soil temperature range during the measurement period: $0.2 \pm 0.03$ °C; mean ± SE). In addition, we minimized any effects diurnal changes in temperature may have on the soil GHG fluxes by randomly selecting the plot to be measured. This means that for every measurement date, the order and sequence in which the plots were sampled was mixed up and which ensured that all the plots had an equal chance of being measured either in the morning or mid-afternoon or late afternoon. To reconfirm that these measures were adequate, and to alleviate any further concerns from the reviewers, we plotted the three soil GHG species fluxes against measurement time (Fig.2). This analysis showed that there is no effect of the measurement time on soil GHG fluxes, and we can confidently say that the negligible diurnal variation in both air and soil temperature did not affect the measured soil GHG fluxes.

[Figure]

Figure 2. Linear relationship between the measured soil greenhouse gas fluxes ($CO_2$, $CH_4$, and $N_2O$) and the time of sampling from the control plots of the NME premised in Budongo Forest Reserve.

We have specifically written in the revised manuscript (in LN 211 to LN 215) that:

*"Soil GHG fluxes were always measured between 9 am and 4 pm throughout the entire study period, while for each measurement day the sequence of plots to be measured was randomly chosen. Together with the very low diurnal variability of air (0.6 ± 0.04 °C; mean ± SE) and soil (0.2 ± 0.03 °C; mean ± SE) temperatures at this tropical forest site, time of measurement of individual gas chambers should, if at all, only have a minimal effect on the measured gas fluxes."*

8. Ln 173 Ammonia or Ammonium?

   Author's response:

   This was a typo. We wrote ammonia in the text but it is supposed to be ammonium. This has been corrected in the revised manuscript. Please see LN 243 for the correction.

9. Ln 264: In Fig 3a it does not appear $CO_2$ fluxes ever went above 250 mg C m$^{-2}$ h$^{-1}$ yet here you give the range as 60 to 330 mg C m$^{-2}$ h$^{-1}$? Please explain or correct.

   Author's response:

   Referring to Fig. 3a was erroneous and an oversight on the part of the authors because the values mentioned in the text were ranges and not means. We have dropped the reference of Fig. 3a.

   The text now reads:

   *"Soil $CO_2$ fluxes varied between 60 and 330 mg C m$^{-2}$ h$^{-1}$ during the measurement period across all treatments." Please see LN 365 to LN 366 of the revised manuscript for the correction."*

10. Ln 288: In Fig 3b, it does not appear $CH_4$ uptake was ever above -200 mg C m$^{-2}$ h$^{-1,}$ yet here you have it as -278 mg C m$^{-2}$ h$^{-1}$? Please explain or correct.

    Author's response:

    Referring to Fig. 3b was equally erroneous and an oversight on the part of the authors because the mentioned values in the text were ranges and not means (presented in Fig. 3b). This has also been corrected in the text. The text now reads:

    *"Across all treatments, phases (transitory and background) and seasons, soil $CH_4$ fluxes varied between an uptake of -278 mg C m$^{-2}$ h$^{-1}$ and a release of 77 mg C m$^{-2}$ h$^{-1}$. Please see LN 391 to LN 392 for the correction."*

11. Ln 353: I think "mirobial" was supposed to be "microbial"

    Author's response:

    This was a typo and has been corrected in the revised version of the manuscript.

    Please see LN 463 for the correction.

12. Ln 400-402: Does the relationship not depend on the form of mineral N (NH4+ of NO3-)? Also, see: Banger, K.; Tian, H.; Lu, C. Do nitrogen fertilizers stimulate or inhibit methane emissions from rice fields? Glob. Chang. Biol. 2012, 18, 3259–3267; for insights on the mechanisms.

    Author's response:

    Yes, whereas it has been shown that the relationship between $CH_4$ uptake and soil mineral N depends on the form of N ($NO_3^-$ or $NH_4^+$), the proportion of $NO_3^-$ and $NH_4^+$ in the soil is key in shaping this relationship. The mineral N fraction at our tropical forest site was dominated by $NO_3^-$ compared to $NH_4^+$, a typical characteristic of open/leaky N terrestrial cycles. As can be seen in Fig. 3 (a, b, and c), we tested the relationship between both $CH_4$ and $NO_3^-$ (Fig. 3a), and $CH_4$ and $NH_4^+$ (Fig. 3b); and found a strong negative relationship between $CH_4$ and $NO_3^-$ (Fig. 3a), but no meaningful relation between $CH_4$ and $NH_4^+$ (Fig. 3b). The correlation between $CH_4$ fluxes and soil mineral N ($NO_3^-$ content plus $NH_4^+$ content) was still negative like it was in the case of $NO_3^-$ (Fig. 3c), simply because $NO_3^-$ ions dominated the soil mineral N fraction at our tropical forest study site. Therefore, despite monitoring soil $NO_3^-$ and $NH_4^+$ on a monthly basis throughout the measurement period, only the soil $NO_3^-$ data set was considered in the analysis because soil $NH_4^+$ was mostly low and often below the detection limit of the reflectometer

at majority of the sampling time points. Most importantly, soil $NH_4^+$ did not have any meaningful correlation to soil $CH_4$ fluxes.

[Figure]

Figure 3. Relationship between $NO_3^-$ (a), $NH_4^+$ (b), mineral N ($NO_3^- + NH_4^+$) (c) and $CH_4$ fluxes

13. Ln 430: What do the results look like when you correlate N2O with either NH4+ or NO3-

Author's response:

There was a relatively weak negative correlation between $N_2O$ and $NO_3^-$ (Fig. 4a), and no correlation at all between $N_2O$ and $NH_4^+$ (Fig. 4b). The correlation between $N_2O$ fluxes and soil mineral N ($NO_3^-$ content plus $NH_4^+$ content) was still negative like it was in the case of $NO_3^-$ (Fig. 4c), simply because $NO_3^-$ ions

dominated the soil mineral N fraction at our tropical forest study site. Soil $NH_4^+$ did not have any meaningful correlation to soil $N_2O$ fluxes.

With respect to comment 12 and 13, we have additionally written in LN 290 to LN 293 of the revised manuscript that:

*"Furthermore, despite monitoring soil $NO_3^-$ and $NH_4^+$ on a monthly basis throughout the measurement period, only the soil $NO_3^-$ data set was used in the analysis because soil $NH_4^+$ was mostly low and often below the detection limit of the reflectometer at majority of the sampling time points."*

[Figure]

Figure 4. Relationship between $NO_3^-$ (a), $NH_4^+$ (b), mineral N ($NO_3^- + NH_4^+$) (c) and $N_2O$ fluxes.

References:

1. Koehler, B., Corre, M. D., Veldkamp, E., Wullaert, H., & Wright, S. J. (2009). Immediate and long-term nitrogen oxide emissions from tropical forest soils exposed to elevated nitrogen input. *Global Change Biology*, *15*(8), 2049-2066.

2. Wright, S. J., Yavitt, J. B., Wurzburger, N., Turner, B. L., Tanner, E. V., Sayer, E. J., ... & Corre, M. D. (2011). Potassium, phosphorus, or nitrogen limit root allocation, tree growth, or litter production in a lowland tropical forest. Ecology, 92(8), 1616-1625.

**Author response to Reviewer_#2**

1. General comments:

1.1. This manuscript presents the results of large-scale nutrient manipulation experiment in a tropical forest in Uganda. Four treatments were considered in this experiment including an unamended control and three different nutrient applications (N, P, and N+P). Greenhouse gas fluxes and other soil data were collected over a fourteen-month experiment. The findings of this manuscript will help advance our understanding of GHG fluxes in African tropical forest ecosystems and how these ecosystems may respond to increases in nitrogen and phosphorus availability.

Author's response:

We appreciate the reviewer's acknowledgement of the tangible contribution our study makes towards the better understanding of the tropical forest responses to changes in ecosystem nutrient dynamics (particularly nitrogen and phosphorus).

1.2. (Part 1 of 4) However, the experimental design is vague and needs additional clarifications.

Author's response:

We appreciate this suggestion. As suggested by both reviewer #2 and the topical editor, we have clarified the experimental design and have written in the revised manuscript as follows (Please see LN 151 to LN 167):

*"The study was conducted within the framework of a running nutrient manipulation experiment (NME). The NME study used a completely randomized design to investigate how the three macronutrients (applied individually as N, P, and potassium (K), and in all possible combinations (N+P, N+K, P+K, N+P+K) as treatments) constrained key ecosystem processes (particularly nutrient cycling, and net primary productivity) in comparison to the unamended control. Each of the eight treatments was replicated four times (hence, n = 32 plots; 8 treatments x 4 replications). While the NME included a K treatment, the soil GHG flux study—the basis for this manuscript was conducted on the N, P and N + P (combination of N and P) plots, and compared to the untreated control plots (n = 16). Only N and P (among nutrient addition plots) were exclusively considered for soil GHG flux measurements because their availability has been shown to limit soil greenhouse gas fluxes from tropical forest biomes. Each treatment plot measured 40 m x 40 m in size but measurements were conducted in the inner measurement core (30 m x 30 m) to avoid boundary effects. A spacing of at least 40 m between experimental plots was ensured to prevent spillover of applied nutrients from the neighboring plots. In order to elicit an ecosystem response, nitrogen was applied at a rate of 125 kg N ha$^{-1}$ yr$^{-1}$ in form of urea ((NH$_2$)$_2$CO), and P at 50 kg P ha$^{-1}$ yr$^{-1}$ as triple superphosphate (Ca(H$_2$PO$_4$)$_2$). The types of fertilizers and application rates used in this study were identical to those used in the Wright et al. (2011) NME. The fertilizer was applied*

*by hand, and in four split dozes every year. Specifically, 31.3 kg N ha$^{-1}$ and 12.5 kg P ha$^{-1}$ were applied to the plots of the NME every three months between May 2018 and June 2020."*

1.2 (Part 2 of 4): The manuscript is also framed as a global change experiment (i.e., increased nutrient deposition), but the amount of N and P applied is not justified and exceeds reasonable nutrient additions in similar ecosystems examining the effects of N and P deposition (e.g., Lu et al. 2018, Van Langehove et al. 2020).

Author's response:

We thank the reviewer for this concern. The reason we applied these rates was to elevate nutrient levels so as to elicit an ecosystem response, and thereby determine how nutrient limitations regulate GHG fluxes. Furthermore, our fertilizer application doses were well within the rates used by other NMEs conducted in the tropics where soil GHG fluxes were measured (Lu et al., 2018 (as mentioned by the reviewer), Koehler et al., 2009, Veldkamp et al., 2013), hence allowing for comparability of findings. Although our study may not qualify as a "realistic" N and P deposition simulation study (because of the high rates), it nevertheless sheds valuable insights on soil GHG flux responses to both abrupt changes in soil nutrient availability (for instance at the onset of the rainy season) and future increases in N and P deposition forecast for this region. Please see also general comment at the beginning of this reply letter.

In the introduction of the revised manuscript, we have expounded on why NMEs use large doses of fertilizers (LN 65 to 70) as follows:

*"One way of understanding how increases in N and P availability (for instance through deposition) affect soil GHG fluxes from tropical forests is through large scale nutrient manipulation experiments (NMEs). NMEs purposely use large doses of N and P (e.g. Cleveland and Townsend, 2006 (150 kg N ha$^{-1}$ yr$^{-1}$ and 150 kg P ha$^{-1}$ yr$^{-1}$), Hall and Matson, 2003 (100 kg N ha$^{-1}$ yr$^{-1}$ and 40 kg P ha$^{-1}$ yr$^{-1}$)) to simulate how future nutrient enrichment of tropical forests (through deposition) could affect soil GHG fluxes (among other ecosystem processes) (Corre et al., 2010)."*

1.2 (Part 3 of 4) I also have concerns about the greenhouse gas sampling frequency and the time between sample collection and measurement in the lab.

Why this greenhouse gas sampling frequency?

This concern, especially regarding N$_2$O fluxes shortly after fertilization was also raised by reviewer #1 (comment 6). Please, see our detailed reply to comment 6 of reviewer #1 above.

The time between sample collection and measurement in the lab.

This concern is addressed in detail in our response to comment 4.9 below, where the reviewer addresses this concern at length.

1.2 (Part 4 of 4) And, in general, the primary findings of the experiment are not effectively placed into the context of global changes and the consequences of increasing reactive nitrogen in the environment. Van Langenhove, L., Verryckt, L.T., Bréchet, L. et al. Atmospheric deposition of elements and its relevance for nutrient budgets of tropical forests. Biogeochemistry 149, 175–193 (2020). https://doi.org/10.1007/s10533-020-00673-8 Lu, X., Vitousek, P. M., Mao, Q., Gilliam, F. S., Luo, Y., Zhou, G., ... & Mo, J. (2018). Plant acclimation to long-term high nitrogen deposition in an N-rich tropical forest. Proceedings of the National Academy of Sciences, 115(20), 5187-5192.

Author's response:

We appreciate this very helpful suggestion by reviewer #2. In light of this suggestion, we have carefully adapted the text in the abstract (LN 19-26), and introduction (LN 57-130) to integrate the global change aspect into our study. Additionally, we have included a section in the discussion (section 4.4.) placing the findings of our study in the context of changing N and P deposition rates in forested ecosystem.

**Specific comments**

2. General abstract comments:

2.1. Consider framing this experiment in the context of global changes, i.e., increased N and P deposition in natural ecosystems. It is not clear from the initial framing if this study concerns managed forests or native forest ecosystems. It is later explained that the experiment occurred in a forest reserve, and this should be clarified for the reader. LN 30: Listing p-values to three significant figures unnecessary. Consider reducing to two significant figures and changing elsewhere in the text.

Author's response:

Although our NME study may not qualify as a "realistic" N and P depositions simulation study, we do agree that it sheds valuable insights on the tropical forest soil GHG flux response to abrupt and sustained increases in N and P availability in these biomes. In light of this recognition, we have carefully adapted the text in the abstract and introduction to integrate the global change aspect into our nutrient manipulation GHG flux study. Also, following the reviewer's advice, we have included upfront (in the abstract) that the study was conducted in a forest reserve, and all the p values used in the text of the revised manuscript have been reduced to two significant values.

3. General introduction comments:

3.1. The impacts of climate change and alterations to the global N and P cycle should be discussed to contextualize this work, particularly in relation to changing N and P dynamics in forested ecosystems. The authors present other NMEs in tropical forests and the lack of experimentation in tropical Africa, but these studies were largely conducted to understand forest responses to N and P deposition. While the authors mention N deposition in LN 96, this global change driver is not presented earlier in the text, and it is an important consideration and rationale for this work.

Author's response:

Kindly see our response to comments 1.2 (part 4 of 4) and 2.1 above.

3.2. LN 96: What about phosphorus? Please provide additional justification for how changes in P deposition could impact tropical forest and GHG budgets.

Author's response:

We have elaborated the effect of increased P deposition or availability on soil GHG fluxes.

Please see LN 90 to LN 102 of the revised manuscript for the elaboration.

3.3. LN 104: Why would P stimulate N release from organic matter? This is mentioned, but not described in detail, in LN 75-84. Perhaps part of my confusion is from the use of organic matter. Do the authors mean soil organic matter or litter? These terms are used interchangeably in LN 81-84.

Author's response:

We appreciate this valuable comment, which helps us to improve clarity of our argument.

We meant to say that P availability has been shown to stimulate increased mineralization of soil organic matter availing more N for soil nitrification or (both) denitrification processes (Mori et al., 2010). We have rephrased this as follows (Please see LN 90 to 91 of the revised manuscript):

*"With respect to P, it has been shown that P availability opens up the N cycle by stimulating soil organic matter mineralization, releasing excess N for soil nitrification or (both) denitrification processes (Mori et al., 2010)."*

4. General methods comments:

4.1. The materials and methods section needs substantial clarifications, including: the rationale for the treatment application rates, when the applications occurred over the course of the experiment, details about the experimental design, and clarification about the GHG flux measurements. Please refer to the detailed comments below.

Author's response:

As suggested by reviewer #2, we have made substantial clarifications on the rationale of the treatment application rates, the timing of the split fertilizer doses, and GHG flux measurements in the text of the methods section of the revised manuscript.

Please see our response to comment 1.2 (part 1-2 of 4).

4.2. LN 113: Please use a more appropriate citation. The authors might consider the WorldClim dataset.

Author response:

The citation has been changed to Lukwago et al. (2020) because their study was conducted in Budongo Forest reserve and they reported long-term temperature (of about 25 °C) and precipitation (of about 1700 mm) values for this region. We used this local reference (Lukwago et al., 2020) as we assume that it gives more realistic data estimates as compared to the global WorldClim dataset that is based on a spatial interpolation with few reliable observations in the region.

4.3. LN 121-125: Additional information about the NME needs to be described. Please add a citation if one exists of previously published work from this site. At a minimum, the text should provide additional clarification regarding the experimental design, i.e., was it randomized? It is also unclear what the number of replicates is in each treatment. Please include in the text that there were four blocks or four replicated plots per treatment.

Author's response:

As suggested by the reviewer, additional information about the NME has been provided in the revised manuscript. Please see our response to comment 1.2 (part 1-2 of 4).

4.4. LN 127-128: The nitrogen and phosphorus additions rates need justification. These rates are unusually high for N and P deposition experiments, and the rates align more closely with those common in agricultural fertilization experiments. This is one of my primary concerns with the framing of this experiment; the applications rates seem far too high to justify as N or P deposition.

Author's response:

We thank the reviewer for this indeed helpful comment. We kindly refer reviewer #2 to our response to comment 1.2 (Part 2 of 4).

4.5. LN 135: How were these soil samples collected, i.e., shovel or core?

Author's response:

For the 0-10 cm depth, soil monoliths (20 cm (L) x 20 cm (W) x 10 cm (D)) were carefully taken from ten different locations within each plot of the NME (n = 32 plots) using a spade. However, for deeper soil layers (0-30 cm and 30- 50 cm), samples were obtained outside the established NME plots in order to minimize modifications to the microenvironment inside the NME plots. Deeper soil sampling was done during a reconnaissance survey conducted at approximately 500 m from the current location of the NME site. During the reconnaissance survey, sixteen plots (n = 16) were established and samples taken from five different locations for every depth interval (i.e. 0-30 cm and 30- 50 cm) using an auger (diameter = 30 mm). As suggested by the reviewer, this information has been included in the revised manuscript, please see LN 173 to LN 185 of the revised manuscript. We have specifically written:

*"For the 0-10 cm depth, soil monoliths (20 cm (L) x 20 cm (W) x 10 cm (D)) were carefully taken from ten different locations within each plot of the NME (n = 32 plots) using a spade. However, for deeper soil layers (0-30 cm and 30- 50 cm), samples were obtained outside the established NME plots in order to minimize modifications to the microenvironment inside the NME plots. Deeper soil sampling was done during a reconnaissance survey conducted at approximately 500 m from the current location of the NME site. During the reconnaissance survey, sixteen plots (n = 16) were established and samples taken from five different locations in each plot for every depth interval (i.e. 0-30 cm and 30- 50 cm) using an auger (diameter = 30 mm). The samples from each depth were mixed thoroughly in a basin, and about 500 g of the homogenized samples sent to the soils laboratory of the University of Göttingen, Germany, for analysis."*

4.6. LN 148: I have concerns regarding this sampling frequency and the subsequent calculations of GHG annual fluxes. This measurement frequency is far too coarse to capture the sensitivity of N2O to precipitation events. From Figure 2, it appears like there were many pulses in precipitation over the experimental period, which may have resulted in substantial N2O release. While I acknowledge the difficulty in sampling at a twice weekly or weekly sampling frequency, the manuscript should describe why this monthly interval was selected for measurement.

Author's response:

We thank reviewer #2 for this comment and refer him/her to our detailed response to comment 6 of reviewer #1 because we explained this concern at length there.

4.7. LN 151-152: Was litter/residue left inside the chamber or was the soil kept bare?

Author's response:

Litter/residue was left inside the chamber. However, the chamber was always maintained vegetation free in order to avoid measuring night respiration of the plants during chamber closure.

We have written in LN 201 to 203 of the revised manuscript as follows:

*"Litter was not removed from the chamber. However, all the chamber bases were always maintained vegetation free throughout the gas sampling period in order to avoid measuring plant night respiration during chamber closure."*

4.8. LN 149-150: I have concerns about the area of the chambers and the sampling times used in this experiment. Carbon dioxide fluxes are usually orders of magnitude greater than N2O or CH4; a larger chamber area is usually necessary to estimate these fluxes from soil. Furthermore, while the sampling times for N2O and CH4 make sense, I am concerned that CO2 may have plateaued during this interval, impacting CO2 diffusion, and the CO2 concentration measured. Did the authors test for a linear

relationship in their pooled and unpooled approach? How representative do the authors feel the chambers were of the overall plot GHG fluxes given the small size of these chambers?

Author's response:

Chamber design:

In light of the expected spatial variability in $N_2O$ and $CH_4$ fluxes (please see Fig. 6), we randomly deployed four chamber bases (with a diameter of 23.7 cm) in every plot, each covering a ground area of about 0.044 $m^2$. Our chamber dimensions are well within the minimum chamber area recommendation by Pavelka et al. (2018), and consistent in design with those used in numerous studies conducted in tropical forests (e.g. Koehler et al. 2009, Matson et al., 2017). With our chamber design, we were able to cover a total ground area of 0.176 $m^2$ per plot (four chambers per plot x 0.044 $m^2$), which in combination with obtaining a composite gas sample from the head airspaces of the four randomly installed chamber bases at each of the four time intervals, allowed us the opportunity to capture the expected variability in $CH_4$ and $N_2O$ fluxes.

Sampling times used in this experiment could result in $CO_2$ saturation:

The chamber closure time adopted for our study closely aligned with those used by Koehler et al. (2009) and Matson et al. (2017) to estimate soil carbon dioxide (among other soil greenhouse gas) fluxes from tropical forest ecosystems, and were well below the 45-minute maximum chamber closure period recommended by Pavelka et al. (2018). All our measurements were also tested for any signs of saturation during the 33 minutes of chamber closure. Overall, we could not find any substantial decrease in $CO_2$ concentration for samples taken at the 33 minute interval. This is exemplarily shown for a randomly selected sub-stet of $CO_2$ measurements (see Fig. 5).

However, we thank the reviewer for the important hint, so we also clarified this for the reader in the manuscript. We have written in LN 206 to LN 208 of the revised manuscript that:

*"The 30 minute maximum chamber closure period used in this study was well under the threshold recommended by Pavelka et al. (2018)), but, comparable to other tropical GHG flux studies (e.g. Corre et al. (2010), Koehler et al. (2009a), Matson et al. (2017))."*

We have also additionally written in LN 236 to LN 238 of the revised manuscript that:

*"As a quality check, the linearity of $CO_2$ increase during chamber closure was inspected by comparing the $CO_2$ concentrations (of each chamber measurement) with time since chamber closure, and thereafter, determined the goodness of fit for the linear regression model (the $R^2$). The $R^2$ for all measurement was $0.992 \pm 0.001$ (mean $\pm$ SE)."*

Linear relationship in their pooled and unpooled approach:

We collected both pooled and unpooled gas samples for the month of February 2020 and tested how the pooled approach compared to the unpooled for all the three gases. There was no significant difference in the fluxes of all the three soil GHGs measured with either the pooled or unpooled approaches (see Fig. 6 below). This is also mentioned in LN 208-210 of the revised manuscript and Fig. 6 has been added as supplementary material in the revised manuscript.

[Figure]

Figure 5. Linear increase in $CO_2$ concentrations with chamber closure time for five randomly selected chamber measurements in each of the five batches.

How representative were the measurements given the size of the chamber used:

As can be seen from the unpooled data of the month of February 2020 (Fig. 6), $CH_4$ and $N_2O$ fluxes exhibited a high spatial variability compared to $CO_2$ fluxes. We took care of the high spatial variability of $CH_4$ and $N_2O$ fluxes by randomly deploying four chamber bases, each covering a minimum ground

area of 0.044 m$^2$, resulting in a total ground area of 0.176 m$^2$ per plot (four chambers x 0.044 m$^2$). We are therefore confident that by randomly deploying four chamber bases per plot coupled with obtaining a pooled sample consisting of the four head airspaces at each time interval enabled us to come up with representative flux measurements for every plot.

[Figure]

Figure 6. Comparison of the soil $CO_2$ fluxes (a) soil $N_2O$ fluxes (b), and soil $CH_4$ fluxes (c) from pooled sampling and the mean of four chamber measurements for the month of February 2020 in Budongo Forest reserve. Rho is the spearman correlation coefficient, and CV is the coefficient of variation. Error bars were derived from standard error of the mean.

4.9. LN 159: The duration between sample collection and measurement needs additional information. How long was the duration between sample collection and measurement? While generally stable for period of days to a couple of weeks, exetainers are not ideal for long-term storage of gas samples, which should ideally be measured immediately (up to 72 hours) after collection. Please describe the care that was taken to ensure there was no degradation to the gas samples over time.

Author's response:

The airtightness of the exetainers depends highly on the type of exetainers and septum used. We purposely selected Labco Exetainers (Labco Limited, Lempeter, UK) with screw-on plastic caps fitted with Labco Grey Chlorobutyl Septum for this study. The research group in which the senior author work(ed) established that these exact Exetainers remain leak-proof for extended periods (up to 6 months; Hassler et al. (2015)) if the Labco Grey Chlorobutyl Septum are used and caps are quarter turned. In the Hassler et al. (2015) paper, they wrote: "*tested these Exetainers for an extended period of sample storage (e.g., up to 6 months) and air transport by storing and transporting standard gases of known concentrations in overpressure, and these Exetainers were proven to be leak proof.*"

In this study, we submitted all the gas samples for analysis at the GC at ETH Zürich, Switzerland, within four months (120 days) from the date of sampling. This was done in five batches across the 14 months of field gas sampling. In an extra step (Fig. 7), we use the sample with the highest (for $CO_2$, $N_2O$) or lowest (for $CH_4$) concentration (taken at 33 minutes) to demonstrate that our sample storage time had a negligible effect on the magnitude of the concentrations for the respective gas species. Sample storage time only explains a negligible percentage of the observed variability in $CO_2$ (6 %), $CH_4$ (1.4 %), and $N_2O$ (1.4 %) concentrations, respectively, based on the $R^2$ values. We not only find the effect of storage time on GHG concentrations very negligible but also equally random since this effect was disproportionate across the three gas species yet they were all drawn/stored in the same exetainer. Therefore, the observed negligible variability will not have had any significant implication on the interpretation of the fluxes and conclusions drawn from this study.

We have stated in LN 215 to LN 221 of the revised manuscript that:

"*All collected gas samples were stored in Labco Exetainers (Labco Limited, Lempeter, UK) with screw-on plastic caps fitted with Labco Grey Chlorobutyl Septum because these exetainers have been demonstrated to remain airtight for periods spanning up to six months (Hassler et al., 2015). Additionally, all the plastic caps were screwed on to the exetainers by hand and "quarter turned" prior to sampling to ensure that they were airtight (Pavelka et al., 2018). All the gas-filled exetainers were always shipped to the Department of Environmental Systems Science, ETH Zürich, Switzerland for analysis using a gas chromatograph (GC; Scion 456-GC Bruker, Germany) within a period of four months from sampling.*"

[Figure]

Figure 7. Relationship between the concentration of the GHG samples species ($CO_2$ (a), $CH_4$ (b), and $N_2O$ (c)) taken at the 33-minute interval and the duration of storage before analysis at the gas chromatography.

LN 187: Please provide a citation for this method.

Author's response:

As suggested, we have provided a citation for the method. Please see LN 256 for the citation.

4.10. LN 190-204: The manuscript should include additional details about a) the frequency of measurements, chamber size, etc. for the trenching experiment, and b) how the authors portioned CO2 to autotrophic and heterotrophic sources and a citation for their methodology.

Author's response:

As suggested by reviewer #2, we have provided additional details about the trenching experiment in the revised manuscript including the citation for methodology used (LN 256).

We have added the following details to the revised manuscript:

Citation of the methodology, we have written (Please LN 255 to LN 256 for the changes):

*"To understand the contribution of autotrophic (root) and heterotrophic (microbial) sources to total soil respiration, a trenching treatment was done in all the plots following the protocol of Wang and Yang (2007)."*

Chamber size for the trenching experiment, we have additionally written (Please LN 269 to LN 270 for the changes):

*"Both the trenched and reference chamber bases had a design (area = 0.044 $m^2$, and volume = about 12 L) identical to the one used in the nutrient manipulation soil GHG flux study."*

Frequency of measurements, we have written (Please LN 272 to LN 275 for the changes):

*"$CO_2$ measurements were conducted monthly for a period of 4 months (starting in November 2019 and ending in February 2020). The selected measurement time window represented the transition between the wet season and the long dry season, allowing us to capture how soil moisture constrained the different soil $CO_2$ efflux sources."*

How heterotrophoic and autotrophic respiration sources were partitioned, we have written (Please LN 279 to LN 281 for the changes): *"Heterotrophic (microbial) respiration was equal to the $CO_2$ effluxes from the trenched chamber while autotrophic (root) respiration was the difference between $CO_2$ effluxes from the reference and trenched chambers."*

4.11. More information about the estimation of root biomass (number of cores, how samples were processed) should also be included, especially because these data are discussed in the results and discussion.

Author's response:

As suggested by the reviewer, the details on estimation of root biomass have been included in the revised version of the manuscript. Please see LN 256 to LN 263 for the changes.

*"Prior to trenching, root biomass distribution with depth was determined in order to establish where most roots were located. Root biomass estimation involved digging three profile pits measuring 1 m (L) x 1 m (W) x 1.1 m (D) at the forest site. In every pit, ten soil monoliths (each measuring: 20 cm (L) x 20 cm (W)) were carefully cut out (using a spade and hoe) following a 10 cm depth interval from the surface down to 1 m. The soil monoliths were thoroughly washed to isolate the roots from the bulk soil. The root samples were oven dried at 60 °C for 48 hours and weighed to determine the root biomass per depth increment. The root biomass for each depth interval was calculated as the mean of the root biomass from the three pits for that interval. It was established that over 90 % of the roots were within the top 0.6 m of the soil profile."*

4.12. LN 212: Is it common to refer to MANOVA as LMEMS?

Author's response:

We thank the reviewer for this question. MANOVAs consider two or more response variables while LMEMs typically consider one response variable. Therefore, it is uncommon to refer to MANOVA as LMEMs.

4.13. LN 219: A description of the interpolation method used to calculate annual GHG fluxes should be described here. I am also confused why the authors present these data but did not do any statistical analyses with them? If these data are included in the results, they should be analyzed statistically.

Author's response:

As suggested, the description of the interpolation method has been included in the methods section of the revised manuscript, and annual fluxes have been statistically analyzed using one way analysis of variance (ANOVA).

We have additionally written in the materials and methods section of the revised manuscript that (Please see LN 318 to LN 319):

*"Annual soil GHG fluxes were estimated through a trapezoidal interpolation on the measured monthly soil GHG fluxes, assuming constant flux rates per day."*

4.14. LN 231: Please include the R packages used in the analyses.

Author's response:

As suggested, the key R packages used in the statistical analyses have been added to the methods section of the revised manuscript (Please see LN 316 to LN 317). As indicated in the original manuscript, statistical analyses were mainly accomplished with linear mixed effects model (LMEMs) and one-way analysis of variance (ANOVA). We used the *'nlme'* and *'car'* packages for LMEMs and one-way ANOVA tests respectively.

We have specifically added a sentence in the revised manuscript stating that (Please see LN 316 to LN 317):

*"All the statistical data analyses were performed using R 3.6.3 (R Development Core Team, 2019) using nlme' and 'car' packages for LMEMs and one-way ANOVA tests, respectively."*

5. General results comments: There are several occurrences in tables and figures where analyses are referenced, but they were not described in the text. This information is more appropriate to include at length in the methods section, and it is inappropriate to only provide as footnotes.

Author's response:

We suppose that this comment was made with respect to the trapezoidal interpolation method for estimation of annual GHG fluxes. The details about this approach have been added to the statistical data analysis section of the revised manuscript (Please see response to comment 4.13).

5.1. Table 1: If the authors present isotope data, they should describe how these data were collected.

Author's response:

As this data is not discussed in the paper, we have opted to remove it from the manuscript. Please see Table 1 in the revised manuscript for this change.

5.2. Figure 2: Why were these climatic data not used to estimate 30-yr mean annual temperature and precipitation? The use of this weather stations should be described in the methods section.

Author's response:

The weather station data presented in our manuscript was only available for the period of the experiment (about 2 years). This data was beneficial to understand how for instance precipitation constrained soil GHG fluxes given its direct control on water filled pore space. For the long-term average of the study region, we cited the Lukwago et al. (2020) study (because they reported long-term temperature and precipitation for this forest site), instead of extrapolating of our 2-year climatic snapshot data to the 30-year climatic average which could be misleading. As suggested by the reviewer, we have written in the materials and methods section that (Please see LN 142 to LN 145):

*"The weather data for the experiment period was obtained from a climatic station installed at Budongo Conservation Field station (2 km northwest of the study site), and was beneficial to understand how precipitation constrained soil greenhouse gas fluxes given its direct control on water filled pore space.."*

6. General discussion comments: I do not find the claim that the ecosystem is "complex" a compelling argument for interpreting the results of the study. The manuscript should omit this language. I also recommend the manuscript include an addition section in the discussion placing the findings of this study in context – how do these results fit into findings of other tropical forest NME and changing N and P deposition rates in forested ecosystems? The broader impact and relevance to the science and policy communities would strengthen the framing of the manuscript.

Author's response:

As suggested by the reviewer, we have omitted the argument on the complexity of the ecosystem in the revised manuscript. Additionally, we have introduced a sub section in the discussion (section 4.4., entitled; "***Implications of increasing N and P deposition rates on soil greenhouse gases from tropical forests***") giving the broader implications of our findings with respect to changing N and P deposition rates over tropical forests.

*"**4.4. Implications of increasing N and P deposition rates on soil greenhouse gases from tropical forests***
*While this experiment was established to investigate how nutrient limitations constrain soil GHG fluxes, it also sheds valuable insights on how anthropogenic nutrient inputs (through deposition) may affect future soil GHG fluxes from African tropical forests and other tropical sites with a similarly strong seasonality, soil, and vegetation characteristics (Table 1). Nutrient depositions are often highest immediately after the onset of the rainy season (Wang et al., 2020), especially due to aerosol deposition following burning activities associated with deforestation during the dry season (Giglio et al., 2006; Roberts et al., 2009). Accordingly, we suspect that the increased N inputs during this short time may yield similar responses to those observed in the transitory period measured at this study site; namely—$N_2O$ flushes when reactive nitrogen enters the soil. Although N additions did not elicit a positive $N_2O$ response during the background period, it is quite likely that our fertilization activities (from year 1 to year 2 of the study) had not gone on for long enough to simulate chronic long-term N additions. A study conducted by Koehler et al. (2009b) in Panama showed that 11 years of chronic N addition significantly increased both transitory and background soil $N_2O$ emissions.*

*In addition, this study shows that future increases in P deposition over tropical forests may significantly boost the $CH_4$ sink capacity of tropical forest soils. Also, it was interesting to observe that the addition of N and P simultaneously resulted in increased $CO_2$ effluxes immediately after fertilization likely suggesting a co-limitation of N and P on soil respiration. This means that future increases in deposition of N and P rich ashes (from biomass burning), might result in significant soil $CO_2$ emissions from these biomes, while*

*it is unclear if this is compensated via an increase in photosynthetic $CO_2$ uptake as indicated by Cernusak et al. (2013). In this context, it has been demonstrated by Barkley et al. (2019) that P derived from biomass burning aerosols is more soluble than the P from dust aerosols, hence, the former would have an immediate impact on ecosystem processes."*

6.1. LN 357: See previous comments about CO2 measurement and sampling frequency concerns.

Author's response:

Please refer to our response to comments 1.2 (part 3 of 4), 4.8, and 4.6.

6.2. LN 433: Please provide additional information about P availability would open the N cycle.

Author's response:

We kindly refer reviewer #2 to response to comment 3.3.

7. General conclusion comments: Please clarify the rationale of this experiment: increased nutrient deposition or fertilization for enhanced forest production? Again, all ecosystems are complex, and this is a weak interpretation of the findings of this study.

Author's response:

As we outlined in our response to the topical editor, the primary objective of this study was to understand the roles nitrogen and phosphorus have in regulating soil GHG fluxes in a humid tropical forest in northwestern Uganda. This experiment was one important aspect of a larger experiment where the overarching aim was to disentangle how nutrients regulate ecosystem processes at a hierarchy of scales from microbial communities to ecosystem carbon accumulation (net primary production). To elicit an ecosystem scale response, and to align our results with previous and ongoing nutrient manipulation experiments (NME) across the tropics, we applied relatively high nutrient doses. Nevertheless, we agree with the reviewer #2 that our findings can be extended to shed valuable insights on soil GHG flux responses to both abrupt changes in soil nutrient availability (at the onset of the rainy season) and increases in N and P deposition forecast for this region. It is for this reason, that we have introduced subsection 4.4 to integrate the global change aspect into our study. We have also rephrased the argument on the complexity of the ecosystem not only in section 4.4. but also elsewhere in the discussion and conclusion.

References:

1. Barkley, A. E., Prospero, J. M., Mahowald, N., Hamilton, D. S., Popendorf, K. J., Oehlert, A. M., Pourmand, A., Gatineau, A., Panechou-Pulcherie, K., Blackwelder, P. and Gaston, C. J.: African biomass burning is a substantial source of phosphorus deposition to the Amazon, Tropical Atlantic Ocean, and Southern Ocean, Proc. Natl. Acad. Sci. U. S. A., 116(33), 16216–16221, doi:10.1073/pnas.1906091116, 2019.

2. Cernusak, L.A., Winter, K., Dalling, J.W., Holtum, J.A., Jaramillo, C., Körner, C., Leakey, A.D., Norby, R.J., Poulter, B., Turner, B.L. and Wright, S.J.: Tropical forest responses to increasing atmospheric CO2: current knowledge and opportunities for future research. Funct. Plant Biol., 40, 531-551, doi.org/10.1071/FP12309, 2013.

3. Cleveland, C. C. and Townsend, A. R.: Nutrient additions to a tropical rain forest drive substantial soil carbon dioxide losses to the atmosphere, Proc. Natl. Acad. Sci. U.S.A., 103, 10316-10321, doi:10.1073/pnas.0600989103, 2006.

4. Corre, M. D., Veldkamp, E., Arnold, J., and Wright, S. J.: Impact of elevated N input on soil N cycling

and losses in old-growth lowland and montane forests in Panama. Ecology 91, 1715–1729. doi: 10.1890/09-0274.1, 2010.

5. Giglio, L., Csiszar, I., and Justice, C. O.: Global distribution and seasonality of active fires as observed with Terra and Aqua Moderate Resolution Imaging Spectroradiometers (MODIS) sensors, J. Geophys. Res., 111, G02016, doi:10.1029/2005JG000142, 2006.

6. Hall, S. J., and Matson P. A.: Nutrient status of tropical rain forests influences soil N dynamics after N additions. Ecological Monographs 73:107–129, jstor.org/stable/3100077, 2003.

7. Hassler, E., Corre, M. D., Tjoa, A., Damris, M., Utami, S. R. and Veldkamp, E.: Soil fertility controls soil-atmosphere carbon dioxide and methane fluxes in a tropical landscape converted from lowland forest to rubber and oil palm plantations, Biogeosciences, 12(19), 5831–5852, doi:10.5194/bg-12-5831-2015, 2015.

8. Koehler, B., Corre, M. D., Veldkamp, E. and Sueta, J. P.: Chronic nitrogen addition causes a reduction in soil carbon dioxide efflux during the high stem-growth period in a tropical montane forest but no response from a tropical lowland forest on a decadal time scale, Biogeosciences, 6(12), 2973–2983, doi:10.5194/bg-6-2973-2009, 2009.

9. Lu, X., Vitousek, P. M., Mao, Q., Gilliam, F. S., Luo, Y., Zhou, G., Zou, X., Bai, E., Scanlon, T. M., Hou, E. and Mo, J.: Plant acclimation to long-term high nitrogen deposition in an N-rich tropical forest, Proc. Natl. Acad. Sci. U. S. A., 115(20), 5187–5192, doi:10.1073/pnas.1720777115, 2018.

10. Lukwago, W., Behangana, M., Mwavu, E. N., & Hughes, D. F. (2020). Effects of selective timber harvest on amphibian species diversity in Budongo forest Reserve, Uganda. Forest Ecology and Management, 458, 117809.

11. Matson, A. L., Corre, M. D., Langs, K. and Veldkamp, E.: Soil trace gas fluxes along orthogonal precipitation and soil fertility gradients in tropical lowland forests of Panama, Biogeosciences, 14(14), 3509–3524, doi:10.5194/bg-14-3509-2017, 2017.

12. Mori, T., Ohta, S., Konda, R., Ishizuka, S. and Wicaksono, A.: Phosphorus limitation on $CO_2$, $N_2O$, and NO emissions from a tropical humid forest soil of South Sumatra, Indonesia, ICEEA 2010 - 2010 Int. Conf. Environ. Eng. Appl. Proc., 18–21, doi:10.1109/ICEEA.2010.5596085, 2010.

13. Pavelka, M., Acosta, M., Kiese, R., Altimir, N., Brümmer, C., Crill, P., Darenova, E., Fuß, R., Gielen, B., Graf, A., Klemedtsson, L., Lohila, A., Longdoz, B., Lindroth, A., Nilsson, M., Jiménez, S. M., Merbold, L., Montagnani, L., Peichl, M., Pihlatie, M., Pumpanen, J., Ortiz, P. S., Silvennoinen, H., Skiba, U., Vestin, P., Weslien, P., Janous, D. and Kutsch, W.: Standardisation of chamber technique for $CO_2$, $N_2O$ and $CH_4$ fluxes measurements from terrestrial ecosystems, Int. Agrophysics, 32(4), 569–587, doi:10.1515/intag-2017-0045, 2018.

14. Roberts, G., Wooster, M.J. and Lagoudakis, E.: Annual and diurnal African biomass burning temporal dynamics. Biogeosciences., 6(5), 849-866, doi.org/10.5194/bg-6-849-2009, 2009.

15. Veldkamp, E., Koehler, B. and Corre, M. D.: Indications of nitrogen-limited methane uptake in tropical forest soils, Biogeosciences, 10(8), 5367–5379, doi:10.5194/bg-10-5367-2013, 2013.

16. Wang, Z., Zhang, X., Liu, L., Cheng, M. and Xu, J.: Spatial and seasonal patterns of atmospheric nitrogen deposition in North China, Atmospheric Sci. Lett., 13(3), 188-194, doi.org/10.1080/16742834.2019.1701385, 2020.

17. Wright, S. J., Yavitt, J. B., Wurzburger, N., Turner, B. I., Tanner, E. V. J., Sayer, E. J., Santiago, L. S.,

Kaspari, M., Hedin, L. O., Harms, K. E., Garcia, M. N. and Corre, M. D.: Potassium, phosphorus, or nitrogen limit root allocation, tree growth, or litter production in a lowland tropical forest, Ecology, 92(8), 1616–1625, doi:10.1890/10-1558.1, 2011.

---

## Author Response (AR2)

**Comments to the Author:**

Topical Editor Decision: Publish subject to minor revisions (review by editor)

I am happy with the revisions made by the authors. Some minor edits to consider in the abstract. I find the use of the word 'rare' on line 20 inappropriate...maybe it's semantics. On line 24, the authors define N and P, which were previously mentioned before being defined. Additionally, other chemical compounds e.g. $CH_4$ and $N_2O$ are mentioned without defining. I suggest you define all, or assume that the readership of the journal do not need those to be defined.

**Author response to the Topical Editor**

Dear Dr. Pauline Chivenge,

We are deeply appreciative for your time and valuable suggestions towards improvement of our manuscript (soil-2020-94). We have addressed all the comments highlighted in your communication above. We kindly invite you to see lines 20-21 and 24-26 of the revised manuscript version (with track changes turned on).

Please do not hesitate to contact me in case of any further questions or information you may need about our manuscript.

Yours sincerely,

Joseph Tamale